# Heterozygous frameshift variants in *HNRNPA2B1* cause early-onset oculopharyngeal muscular dystrophy

Hong Joo Kim [1,39], Payam Mohassel [2,39], Sandra Donkervoort[2], Lin Guo[3,4], Kevin O'Donovan[1], Maura Coughlin[1], Xaviere Lornage[5], Nicola Foulds[6], Simon R. Hammans[7], A. Reghan Foley[2], Charlotte M. Fare [3], Alice F. Ford[3], Masashi Ogasawara[8,9], Aki Sato[10], Aritoshi Iida [9], Pinki Munot[11], Gautam Ambegaonkar[12], Rahul Phadke[13], Dominic G. O'Donovan[14], Rebecca Buchert [15], Mona Grimmel[15], Ana Töpf[16], Irina T. Zaharieva[11], Lauren Brady[17], Ying Hu[2], Thomas E. Lloyd [18], Andrea Klein[19,20], Maja Steinlin[19], Alice Kuster[21], Sandra Mercier[22,23], Pascale Marcorelles[24], Yann Péréon[25], Emmanuelle Fleurence [26], Adnan Manzur[11], Sarah Ennis [27], Rosanna Upstill-Goddard [27], Luca Bello [28], Cinzia Bertolin[29], Elena Pegoraro[28], Leonardo Salviati [30], Courtney E. French [31], Andriy Shatillo [32], F. Lucy Raymond [33], Tobias B. Haack[15], Susana Quijano-Roy[34], Johann Böhm[5], Isabelle Nelson[35], Tanya Stojkovic[36], Teresinha Evangelista[37], Volker Straub [16], Norma B. Romero[36,37], Jocelyn Laporte [5], Francesco Muntoni[11], Ichizo Nishino [8,9], Mark A. Tarnopolsky [17], James Shorter[3], Carsten G. Bönnemann [2✉] & J. Paul Taylor [1,38✉]

Missense variants in RNA-binding proteins (RBPs) underlie a spectrum of disease phenotypes, including amyotrophic lateral sclerosis, frontotemporal dementia, and inclusion body myopathy. Here, we present ten independent families with a severe, progressive muscular dystrophy, reminiscent of oculopharyngeal muscular dystrophy (OPMD) but of much earlier onset, caused by heterozygous frameshift variants in the RBP hnRNPA2/B1. All disease-causing frameshift mutations abolish the native stop codon and extend the reading frame, creating novel transcripts that escape nonsense-mediated decay and are translated to produce hnRNPA2/B1 protein with the same neomorphic C-terminal sequence. In contrast to previously reported disease-causing missense variants in *HNRNPA2B1*, these frameshift variants do not increase the propensity of hnRNPA2 protein to fibrillize. Rather, the frameshift variants have reduced affinity for the nuclear import receptor karyopherin β2, resulting in cytoplasmic accumulation of hnRNPA2 protein in cells and in animal models that recapitulate the human pathology. Thus, we expand the phenotypes associated with *HNRNPA2B1* to include an early-onset form of OPMD caused by frameshift variants that alter its nucleocytoplasmic transport dynamics.

A full list of author affiliations appears at the end of the paper.

RNA-binding proteins (RBPs) play central roles in regulating RNA metabolism, including transcription, RNA splicing, polyadenylation, stabilization, localization, and translation[1]. In eukaryotic cells, RBPs are frequently associated with RNA in ribonucleoprotein (RNP) granules, which are complex assemblies that arise via liquid–liquid phase separation (LLPS)[2]. Many RBPs contain intrinsically disordered low-complexity domains (LCDs) that support LLPS through weak, multivalent interactions. The ability of RBPs to facilitate LLPS and partition RNAs into condensates is an important mechanism by which they govern RNA metabolism[3].

Pathogenic missense variants in RBPs such as TDP-43[4,5], hnRNPA1[6–9], hnRNPA2/B1[6,10], hnRNPDL[11–14], FUS[15,16], and TIA1[17,18] cause a spectrum of diseases with pleomorphic phenotypic manifestations including amyotrophic lateral sclerosis (ALS)/motor neuron disease, frontotemporal dementia (FTD), inclusion body myopathy (IBM), distal myopathy, and Paget's disease of bone (PDB). Patients may also present with complex, combined phenotypes impacting muscle, brain, and bone, a syndrome that has been termed multisystem proteinopathy (MSP)[19]. Remarkably, most pathogenic variants in RBPs are located in intrinsically disordered LCDs[4–6,15–18]. Studies using purified proteins and cellular expression models have demonstrated that LCD variants can alter LLPS and ultimately result in aggregation and fibrillization of the mutant protein[6,9]. We have proposed that pathological phase separations of this sort can drive neuronal dysfunction or demise either by altering the function of biomolecular condensates, by producing toxic, aggregated protein, or both[20,21].

We previously reported a family with the MSP phenotype manifesting as degeneration of muscle, bone, or brain, either alone or in combination in different family members, caused by a rare missense mutation in *HNRNPA2B1* [NM_002137, p.(D290V)][6]. This mutation, which is within the LCD of the protein, promotes assembly of hnRNPA2 into self-seeding fibrils by introducing a potent steric-zipper motif, thereby dysregulating polymerization and ultimately driving the formation of cytoplasmic inclusions[6,22]. A second heterozygous missense mutation in *HNRNPA2B1* [NM_002137, p.(P298L)], also within the LCD and predicted to promote aggregation, was found to cause pure PDB without additional multisystemic features[10]. MSP-causing heterozygous missense mutations have been identified in the LCD of two additional RBPs, hnRNPA1[6] and TIA1[18]. Moreover, ALS/FTD-causing mutations in the RBP TDP-43 most often impact the LCD[23], and ALS-causing mutations of the RBP FUS sometimes impact the LCD[24]. Similar to the observed consequences of missense mutations to the LCD of hnRNPA2/B1, many of the disease-causing mutations in the LCDs of hnRNPA1, hnRNPDL, TIA1, TDP-43, and FUS have been shown or are predicted to promote assembly of self-seeding fibrils, ultimately driving the formation of cytoplasmic inclusions[6,18,25–28].

Modest expansion of a polyalanine repeat tract in *PABPN1*, which also encodes an RBP, underlies the majority of cases of oculopharyngeal muscular dystrophy (OPMD), a late-onset (typically 5th decade) disease characterized by ptosis and dysphagia, and in rare cases progressive limb weakness and later stage vertical ophthalmoparesis[29]. OPMD is clinically distinct from the pleomorphic syndrome MSP, and, unlike other RBP-associated diseases, has a homogeneous and recognizable phenotypic spectrum that predominantly manifests in specific skeletal muscles.

Here, we report and characterize 10 independent families with a severe, progressive, early-onset OPMD-like phenotype (eoOPMD) that is distinct from MSP and is caused by a novel class of heterozygous frameshift variants in the LCD of hnRNPA2/B1. In contrast to the missense variants that cause MSP or PDB, these frameshift variants do not promote protein fibrillization. Rather, these frameshift variants alter the C-terminal portion of the nuclear localization sequence (NLS) of hnRNPA2 and promote cytoplasmic accumulation of the protein by impairing its interaction with the nuclear import receptor karyopherin β2 (Kapβ2). Furthermore, our findings demonstrate that two classes of disease-causing mutation have distinct consequences for the hnRNPA2 protein: the first (missense mutations in the LCD) increases the propensity toward fibrillization, whereas the second (frameshift mutations that partially impair nuclear import) increases the cytoplasmic concentration of the protein without changing its intrinsic propensity to fibrillize. Both classes of mutation culminate in muscle pathology, albeit with different clinical syndromes characterized by differing ages of onset and anatomic distribution of pathology.

## Results

**HNRNPA2B1 frameshift variants manifest with a distinct, early-onset progressive myopathy.** We identified 11 patients from 10 independent families with progressive early-onset myopathy with ophthalmoplegia, ptosis, and respiratory insufficiency of variable degrees (Table 1, Fig. 1a). Six patients presented with first recognition of symptoms before 2 years of age, ranging from respiratory insufficiency requiring tracheostomy at birth, to delayed motor milestones or isolated ptosis and ophthalmoplegia. All patients had axial weakness and progressive proximal and distal weakness, which was more pronounced in the lower extremities compared to the upper extremities (Fig. 1b). Respiratory involvement was variable, ranging from a severely decreased forced vital capacity (FVC) (21% predicted) to normal. Ptosis and progressive ophthalmoparesis were uniformly present in all patients and were noted as early as 6 months of age. Cognition was normal and patients did not have a history of seizures, cardiac involvement, or bone abnormalities. Serum creatine kinase levels were elevated in all patients, ranging from 3 to 35 times the upper limit of normal (Table 1). Detailed clinical and genetic information for each patient can be found in Supplementary Table 1 and Supplementary Information.

To elucidate the genetic origin of the muscle disease, we pursued exome sequencing and identified nine (one recurrent) distinct heterozygous frameshift variants in *HNRNPA2B1* (MIM:600124), which were absent in the Genome Aggregation Database (gnomAD)[30] (Fig. 1c and Supplementary Table 1). In family 4, the disease was inherited in a dominant fashion and the variant segregated with disease. The disease occurred sporadically in the remaining nine families (Fig. 1a) and the *HNRNPA2B1* variants were confirmed to be de novo (i.e., absent in unaffected parents) in seven families for which parental samples were available for testing.

**Muscle imaging reveals myopathic changes with discrete foci of increased T1 signal.** Muscle MRI images of lower extremities showed muscle atrophy and a heterogenous pattern of T1 signal hyperintensity, suggestive of structural changes in the muscle composition (Fig. 1d). Presence of patchy foci of T1 signal hyperintensity, suggestive of focal fatty replacement of muscle, was noted in mildly affected muscles, while more severely affected muscles showed a diffuse pattern of T1 hyperintensity. In the anterior thigh compartment, the rectus femoris muscle was relatively spared. Medial and posterior thigh compartments were selectively affected, albeit not uniformly, with the gracilis muscle appearing relatively spared. In the lower legs, the peroneus group, soleus, and lateral gastrocnemius muscles were selectively affected; however, the appearance of this pattern was variable among patients. In the head and neck MRI images, T1 hyperintensity in the tongue was notable (Fig. 1d, arrows).

**Table 1 Clinical features of patients with pathogenic variants in *HNRNPA2B1*.**

| Family/Patient | Ethnicity | Sex/age at last examination (y) | Pathogenic *HNRNPA2B1* variant[a] | Inheritance | Onset | Pattern of muscle weakness | CK level U/L |
|---|---|---|---|---|---|---|---|
| F1/P1 | Italian/Colombian | M/12 | c.992delG, p.(G331Efs*28) | de novo | Swallowing difficulties (2 y) | Ptosis, ophthalmoplegia, tongue weakness, dysphonia, symmetric proximal and distal weakness in LE > UE; moderate progression, dysphagia, respiratory insufficiency | 724 |
| F2/P2 | Northern European | M/17 | c.981delA, p.(G328Afs*31) | de novo | Neonatal feeding difficulties | Ptosis, ophthalmoplegia, symmetric proximal and distal weakness in LE > UE; moderate progression, dysphagia | 602 |
| F3/P3 | Ukrainian | F/17 | c.984delC, p.(S329Vfs*30) | de novo | Muscle weakness (6 y) | Ptosis, ophthalmoplegia, dysphonia, symmetric proximal and distal weakness in LE > UE; dysphagia, moderate progression | 994 |
| F4/P4 | English | F/40 | c.966delA, p.(N323Tfs*36) | Dominant | Ptosis, ophthalmoplegia (18 y) | Ptosis, ophthalmoplegia, symmetric proximal and distal weakness in LE > UE; moderate progression, dysphagia, respiratory insufficiency | 808 |
| F4/P5 | English | F/43 | c.966delA, p.(N323Tfs*36) | Dominant | Ptosis, ophthalmoplegia, dysphagia (17 y) | Ptosis, ophthalmoplegia, symmetric proximal and distal weakness in LE > UE; moderate progression, dysphagia, respiratory insufficiency | 1105 |
| F5/P6 | English | F/12 | c.1001delG, p.(G334Vfs*25) | Presumed de novo | Respiratory insufficiency requiring tracheostomy (birth) | Ptosis, ophthalmoplegia, asymmetric proximal and distal weakness LE > UE; mild progression, dysphagia, respiratory insufficiency | 770 |
| F6/P7 | French | F/9 | c.996_997dupTG, p.(G333Vfs*27) | de novo | Ptosis, ophthalmoplegia (6 months) | Ptosis, ophthalmoplegia, dysphonia, dysphagia, axial weakness; respiratory insufficiency, dysphagia, loss of independent ambulation (8 years) | 536 |
| F7/P8 | French | M/20 | c.980_988del, p.(G327Vfs*30) | de novo | Delayed motor milestones (infancy) | Ptosis, ophthalmoplegia, dysphonia, axial weakness, symmetric proximal UE and LE weakness, respiratory insufficiency | 2484 |
| F8/P9 | Italian | M/35 | c.974delG, p.(G325Vfs*34) | de novo | Slower than peers (5 y) | Ptosis, ophthalmoplegia, tongue weakness, symmetric proximal and distal weakness LE > UE; moderate/severe progression, respiratory insufficiency, dysphagia | 7067 |
| F9/P10 | Indian | F/7 | c.996_997dupTG, p.(G333Vfs*27) | de novo | Respiratory illness with acute onset ptosis (18 months) | Ptosis, ophthalmoplegia, dysphonia, symmetric LE > UE proximal, distal and axial weakness; loss of independent ambulation (7 years), ankle contractures, dysphagia, respiratory insufficiency | 424 |
| F10/P11 | Japanese | F/18 | c.1001_1002dupGT, p.(Y335Vfs*25) | Presumed de novo | Slow runner (childhood) | Ptosis, ophthalmoplegia, symmetric LE > UE proximal muscle weakness, respiratory insufficiency | 501 |

*M* male, *F* female, *y* years, *CK* creatine kinase, *LE* lower extremity, *UE* upper extremity.
[a]Transcript ID: NM_002137.

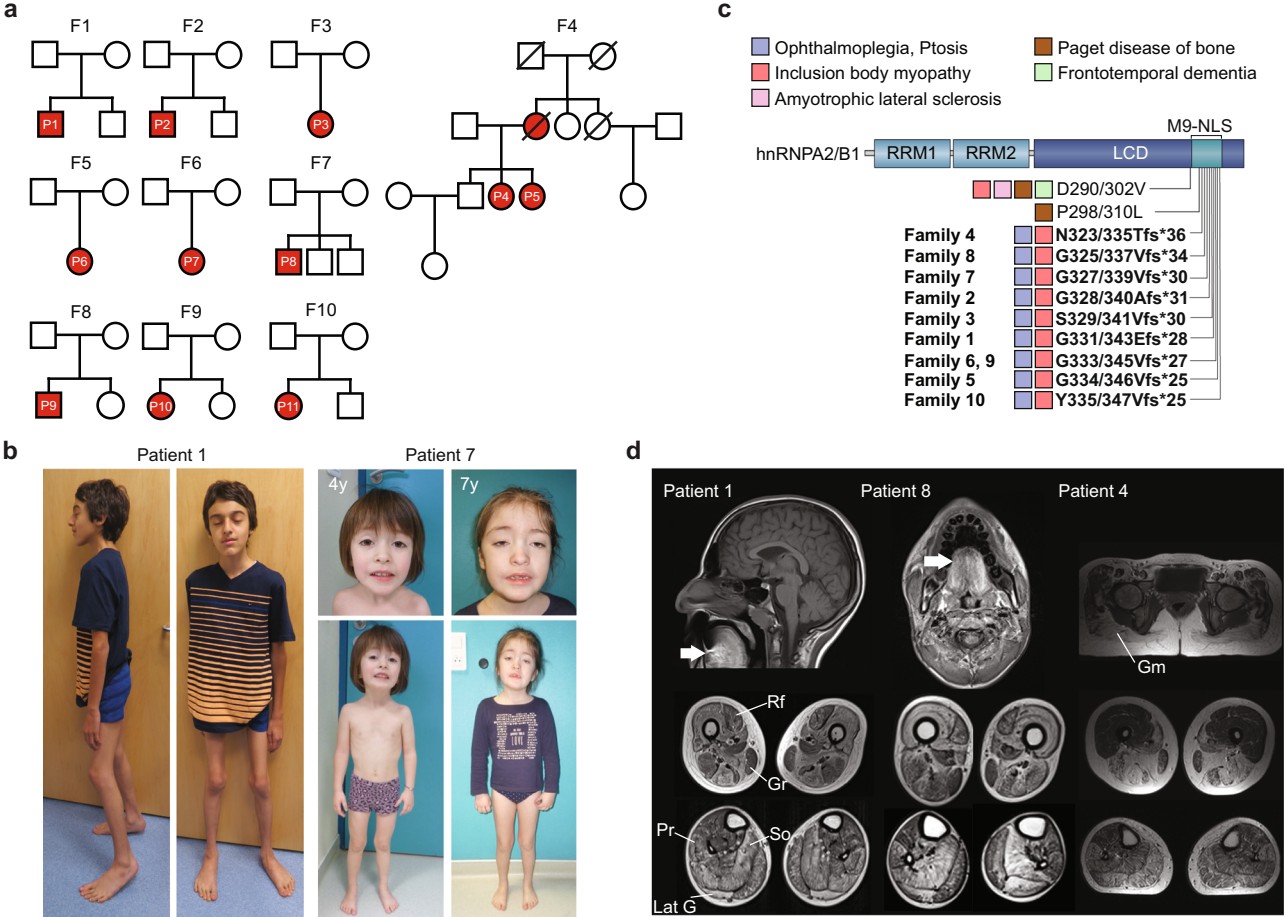

**Fig. 1 Clinical characteristics and genetic findings in ten independent families with *HNRNPA2B1* variants. a** Pedigrees of the 10 families. F family, P patient. Filled red indicates affected individuals with eoOPMD phenotype and heterozygous *HNRNPA2B1* frameshift variant. **b** Patient 1 with c.992delG, p.(G331Efs*28) variant, and patient 7 with c.996_997dupTG, p.(G333Vfs*27) showing prominent ptosis and muscle atrophy. Note the progression of ptosis in patient 7 from 4 years of age to 7 years of age. **c** hnRNPA2/B1 domain structure with conserved regions. Variants identified from previous studies and in this study (in bold) are indicated in both hnRNPA2 and hnRNPB1 isoforms with their associated phenotypic features. RRM RNA-recognition motif, LCD low complexity domain, M9-NLS M9-nuclear localization signal. **d** T1 MRI images of the head and lower extremities. Head and neck MRI highlights T1 hyperintensity in the tongue (arrows indicate "bright tongue sign"). In the hips, gluteus maximus (Gm) is selectively affected. In the lower extremities, there is selective involvement of the posterior and medial thigh compartment with relative sparing of the rectus femoris (Rf) and gracilis (Gr) muscle. Lower leg images show relative sparing of the tibialis anterior and selective involvement of the peroneus group (Pr), soleus (So), and lateral gastrocnemius (Lat G).

***HNRNPA2B1* frameshift variants manifest pathologically with rimmed vacuoles and intracytoplasmic aggregates**. Muscle biopsies of the patients were consistent with a chronic degenerative myopathy, characterized by myofiber atrophy, fiber size variability, increased internalized nuclei, and presence of subsarcolemmal and cytoplasmic rimmed vacuoles without inflammatory infiltrates (Fig. 2a, b). Light microscopy revealed that the vacuoles contained proteinaceous material (Fig. 2c–f), and ultrastructural analysis confirmed the presence of membrane-bound vacuoles containing debris in a "myelin-like" configuration often attributed to excess autophagy (Fig. 2g, h). Cytoplasmic, perinuclear, and on occasion intranuclear tubulofilamentous inclusions with ~15–20 nm thickness were also identified, indicating the accumulation of microfibrillar protein aggregates (Fig. 2i, j, arrows and inset). Immunofluorescence staining of muscle tissue highlighted scattered myofibers with hnRNPA2/B1-positive inclusions that partially colocalized with P62, ubiquilin 2, TIA1, and TDP-43, and to a lesser degree with ubiquitin (Fig. 2k–o), similar to the end-stage fibrillar accumulation of RBPs in microfibrillar structures observed in related disorders[6]. These inclusions did not stain positive for Z-disk-associated

proteins that typically aggregate in myofibrillar myopathies (Supplementary Fig. 1) and were unvested within membrane layers.

**Frameshift variants cluster in the hnRNPA2B1 LCD and escape RNA quality control degradation**. *HNRNPA2B1* is expressed as two alternatively spliced isoforms, *HNRNPA2* (NM_002137, 11 exons) and *HNRNPB1* (NM_031243, 12 exons). The shorter isoform, *HNRNPA2*, which lacks an exon (exon 2 of *HNRNPB1*) and its associated 12 amino acids in the N-terminal region, is the main isoform, accounting for 90% of the protein in most tissues. All of the frameshift variants we identified in our cohort cluster in exon 10 of *HNRNPA2* and exon 11 of *HNRNPB1*, which encode the highly conserved LCD and are respectively the last coding exon in each isoform. These frameshifts all abolish the native stop codon and extend the reading frame (Fig. 1c). Irrespective of the point at which each frameshift occurred, all mutations result in the same frameshift with a common C-terminal sequence (VMVGGADTELLPICHGLH-CINRRG) (Supplementary Fig. 2a, b). RT-PCR analysis in muscle tissue from patients 1 and 2 suggested that these frameshift

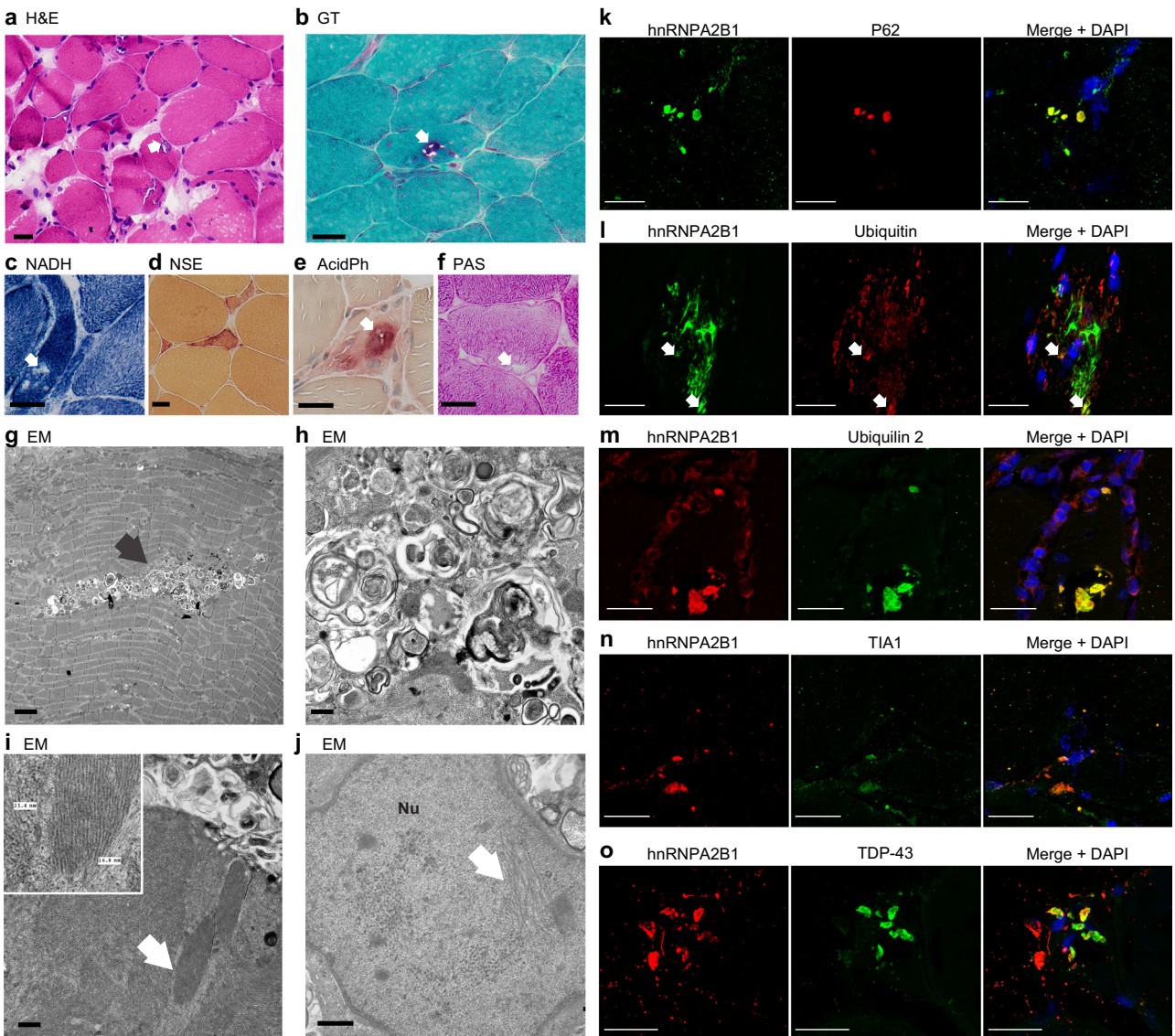

**Fig. 2 Histological findings of hnRNPA2B1 myopathy. a** Hematoxylin and eosin (H&E) image of right rectus femoris muscle biopsy obtained at 8 years of age from patient 1 with c.992delG, p.(G331Efs*28) showing myofiber atrophy, fiber size variability, and internalized nuclei with several fibers with rimmed subsarcolemmal vacuoles (arrow). **b** Modified Gömöri trichrome (GT) staining highlights similar findings and rimmed vacuoles (arrow) (patient 1 muscle biopsy). **c–f** Vacuole contents (arrow) do not stain positive with NADH, suggesting they are devoid of mitochondria (**c**) and do not show increased non-specific esterase (NSE) activity (**d**). The vacuoles (arrow) do not contain glycoproteins based on periodic acid–Schiff (PAS) stain (**f**) but have increased acid phosphatase (AcidPh) activity (arrow) (**e**), suggesting they are lysosomal or autophagic in origin. NSE stain (**d**) also highlights a few angular atrophic fibers, suggestive of mild acute neurogenic atrophy. **c–f** are from patient 2 muscle biopsy. **g, h** Electron microscopy (EM) studies showing marked autophagic changes (black arrow) and vacuoles containing membranous myelin-like whorls. **i, j** Many myofibers contain areas with ~15–20-nm thick, tubulofilamentous inclusions (**i**, white arrow and inset), which on occasion were also seen within the nuclei (Nu) near the vacuoles (j, white arrow). **g–i** are from patient 2 muscle biopsy, **j** is from patient 8 muscle biopsy. **k–o** Immunofluorescence staining of muscle biopsy of patient 1 with c.992delG, p.(G331Efs*28) showing cytoplasmic hnRNPA2/B1-positive inclusions that partially co-localize with P62 (**k**), ubiquilin 2 (**m**), TIA1 (**n**), and TDP-43 (**o**). Co-localization with ubiquitin (**l**) is restricted to very few perinuclear and cytoplasmic puncta (arrows). Scale bars: **a–f** 25 μm; **g** 2 μm; **h–j** 500 nm; **k–o** 20 μm. The micrographs shown are representative images of a single diagnostic muscle biopsy in each indicated patient. No independent replicates were performed.

variants escape degradation by the RNA quality control system (Supplementary Fig. 2c).

**Frameshift variants alter the nucleocytoplasmic ratio of hnRNPA2 and enhance its recruitment to RNP granules.** Nucleocytoplasmic shuttling of hnRNPA2 is regulated by its 40-amino acid M9 sequence located within the C-terminal LCD (Fig. 3a, b). The M9 sequence serves as both an NLS and a nuclear export signal (NES) and is recognized by the nuclear transport receptor Kapβ2 (also known as transportin 1, TNPO1)[31–33]. M9-NLSs are structurally disordered and have an overall basic character with weakly conserved sequence motifs composed of an N-terminal hydrophobic or basic motif followed by a C-terminal R/H/KX$_{2-5}$PY consensus sequence and its flanking region[34] (Fig. 3b). Due to the strict conservation of proline-tyrosine (PY) residues and the importance of these residues in the binding of Kapβ2, M9-NLSs are now often defined as PY-NLSs, a more

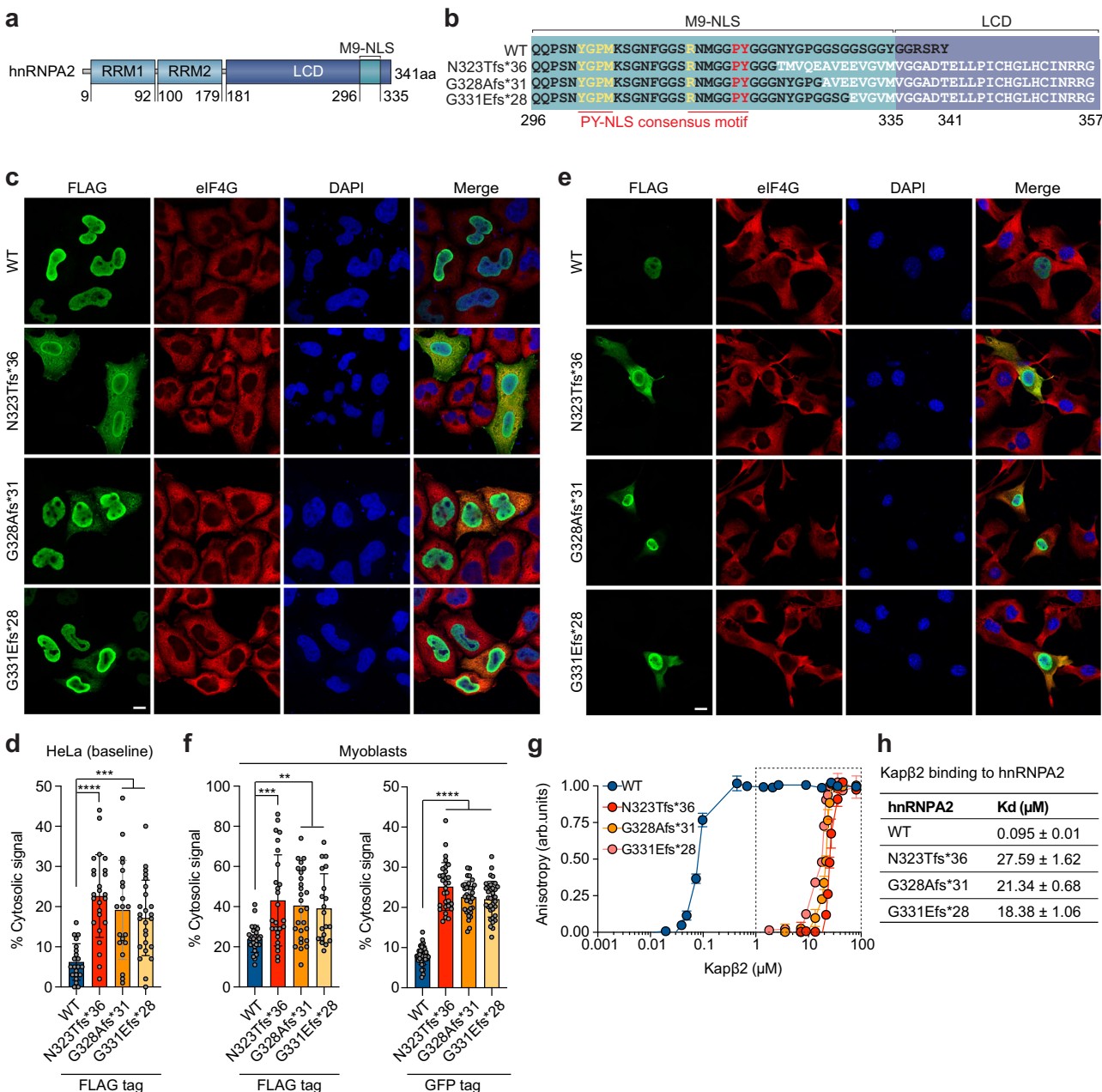

**Fig. 3 Frameshift mutations impair nucleocytoplasmic trafficking of hnRNPA2 by disrupting interaction between hnRNPA2 and Kapβ2. a** Domain architecture of hnRNPA2 illustrating RNA recognition motifs 1 and 2 (RRM1 and RRM2), a low complexity domain (LCD), and an M9 nuclear localization signal (M9-NLS) within the LCD. **b** Amino acid sequences of relevant domains in WT hnRNPA2 and frameshift mutants. Consensus PY-NLS motifs within the M9-NLS are underlined in red. **c** Intracellular localization of indicated FLAG-tagged hnRNPA2 proteins under basal conditions in HeLa cells. eIF4G was used as a cytoplasmic and stress granule marker. Scale bar, 10 μm. **d** Quantification of hnRNPA2 cytosolic signal intensity in HeLa cells as shown in (**c**). An interleaved scatter plot with individual data points is shown; error bars represent mean ± s.d. For WT, N323Tfs*36, G328Afs*31, and G331Efs*28, $n = 21$, 25, 20, and 25 cells, respectively. ***$P = 0.0001$ (WT vs. G328Afs*31), ***$P = 0.0006$ (WT vs. G331Efs*28), and ****$P < 0.0001$ by one-way ANOVA with Dunnett's multiple comparisons test. **e** Intracellular localization of indicated FLAG-tagged hnRNPA2 proteins in C2C12 myoblasts. eIF4G was used as a cytoplasmic and stress granule marker. Scale bar, 10 μm. **f** Quantification of FLAG-tagged or GFP-tagged hnRNPA2 cytosolic signal intensity in C2C12 myoblasts. An interleaved scatter plot with individual data points is shown; error bars represent mean ± s.d. For FLAG-hnRNPA2, $n = 26$, 26, 27, and 20 cells for WT, N323Tfs*36, G328Afs*31, and G331Efs*28, respectively. For GFP-hnRNPA2, $n = 36$, 30, 33, and 35 cells for WT, N323Tfs*36, G328Afs*31, and G331Efs*28, respectively. ****$P < 0.0001$, ***$P = 0.0003$ (WT vs. N323Tfs*36), **$P = 0.0017$ (WT vs. G328Afs*31), and **$P = 0.0094$ (WT vs. G331Efs*28) by one-way ANOVA with Dunnett's multiple comparisons test. **g** Fluorescence polarization measurements between TAMRA-labeled M9-NLS of hnRNPA2 WT, N323Tfs*36, G328Afs*31, or G331Efs*28 peptide (100 nM) with increasing concentrations of Kapβ2. Peptide sequences are shown in (**b**). Values represent means ± s.e.m. ($n = 3$ independent experiments). **h** $K_d$ values between TAMRA-labeled M9-NLS of hnRNPA2 peptides and Kapβ2 WT. Values were calculated as detailed in Methods and represent means ± s.d. from three independent experiments.

minimal class of NLS that ends on the PY motif[34] (Fig. 3b). Because hnRNPB1 is a much less abundant isoform that uses precisely the same mechanism of nuclear import and is expected to be impacted in the same way as hnRNPA2, it will not be discussed further.

All nine *HNRNPA2B1* frameshift variants identified in our cohort altered the M9-NLS amino acid sequence by shifting the reading frame by one base pair (Supplementary Fig. 2a, b), suggesting that nucleocytoplasmic trafficking might be impaired by these variants. To examine the effect of these C-terminal frameshift variants on nucleocytoplasmic trafficking of hnRNPA2, we expressed FLAG-tagged versions of wild-type (WT) and three different mutant hnRNPA2 proteins (N323Tfs*36, G328Afs*31, and G331Efs*28) in HeLa cells. Whereas hnRNPA2 WT almost exclusively localized to nuclei, N323Tfs*36, G328Afs*31, and G331Efs*28 mutants showed cytoplasmic accumulation at baseline, suggesting impairment of nucleocytoplasmic transport (Fig. 3c, d). The cytoplasmic accumulation of mutant proteins became more apparent when cells were subjected to oxidative stress, which induces assembly of stress granules (Supplementary Fig. 3a, b). We also evaluated the impact of the variants on hnRNPA2 localization in a disease-relevant cell type and found that, similar to HeLa cells, C2C12 myoblast cells (Fig. 3e, f) and differentiated myotubes (Supplementary Fig. 3c) expressing either FLAG-tagged or GFP-tagged hnRNPA2 showed increased cytoplasmic localization of mutant hnRNPA2 proteins compared to WT.

To more clearly demonstrate the distribution of hnRNPA2 variants in cells, we next measured the signal intensities of WT, the MSP-associated D290V mutant, and the eoOPMD-associated N323Tfs*36 mutant hnRNPA2 in the nucleus and cytoplasm. Line scan intensity graphs indicated that both WT and D290V predominantly localized to the nucleus before stress (Supplementary Fig. 4a). Upon oxidative stress (500 μM sodium arsenite), D290V, but not WT, showed association with stress granules, as demonstrated by colocalization with eIF4G punctae (Supplementary Fig. 4b). However, the majority of the hnRNPA2 signal was still found in the nucleus. In contrast, the N323Tfs*36 variant showed substantial accumulation in the cytoplasm before stress (Supplementary Fig. 4a) and strong association with stress granules upon oxidative stress (Supplementary Fig. 4b). When not associated with stress granules, the cytoplasmic distribution of N323Tfs*36 was similar to that of eIF4G, which was diffuse in the cytoplasm, demonstrating that N323Tfs*36 protein remains diffuse rather than in aggregates.

We next introduced fluorescent C-terminal tags to WT and frameshift hnRNPA2 mutants to characterize their respective distributions and to determine whether the C termini of frameshift variants might be cleaved from the full-length protein. In C2C12 cells, C-terminally GFP-tagged hnRNPA2 proteins showed equivalent localization patterns to N-terminally GFP-tagged hnRNPA2 proteins, with WT protein in the nucleus and frameshift mutant proteins showing accumulation in the cytoplasm (Supplementary Fig. 4c). We observed no cytoplasmic aggregation of C-terminally-tagged frameshift mutant proteins, consistent with our observations using N-terminally tagged mutant proteins in HeLa and C2C12 cells (Fig. 3c–f). Furthermore, the C-terminally GFP-tagged hnRNPA2 proteins migrated on a gel with an observed molecular weight of ~62 kDa, which is close to the estimated molecular weight of GFP-tagged full-length protein (hnRNPA2 37 kDa + GFP 27 kDa), and no cleaved product was observed by Western blot analyses (Supplementary Fig. 4d). In addition, solubility assays revealed that all three frameshift variants were more RIPA-soluble than WT proteins (Supplementary Fig. 4e). Thus, we conclude that frameshift variants cause diffuse cytoplasmic accumulation of hnRNPA2

protein in cells and that frameshift mutant proteins are less intrinsically aggregation-prone compared with WT protein.

Distinct from findings in patient muscle biopsies, we did not observe hnRNPA2-positive protein aggregates in cells within the timeframe of our analyses. Thus, the hnRNPA2/B1-positive inclusions observed in patient muscle biopsies may not represent the immediate consequences of these expressed frameshift variants on protein aggregation and homeostasis, and may instead reflect a pathologic manifestation of the disease process over time.

**Frameshift variants impair the interaction between hnRNPA2 and its nuclear transport receptor Kapβ2.** The conserved consensus motifs of the PY-NLS are located proximal to the region in which the *HNRNPA2B1* reading frame is altered by the frameshift variants (Fig. 3b and Supplementary Fig. 2b). Previous studies of hnRNPA1, a homologous protein of hnRNPA2/B1 that also harbors an M9-NLS, have shown that post-translational modifications of residues that are C-terminal to the conserved PY residues inhibit the interaction of hnRNPA1 with Kapβ2 and impair nuclear import of hnRNPA1, indicating that C-terminal flanking regions can influence PY-NLS activity[34,35]. This prior observation in the closely related hnRNPA1 protein suggests that the cytoplasmic accumulation of frameshift mutant hnRNPA2 might reflect loss of Kapβ2 interaction and subsequent impaired nuclear import. To test the impact of frameshift variants on Kapβ2 binding, we synthesized WT and three frameshift mutant peptides (Fig. 3b) with a 5′ tetramethylrhodamine (TAMRA) label and used fluorescence polarization assays to quantify their interaction with Kapβ2[36]. The hnRNPA2 WT peptide bound Kapβ2 with a $K_d$ of 95 nM (Fig. 3g, h). All three frameshift mutant peptides showed a significant decrease in Kapβ2 binding (Fig. 3g, h), suggesting that frameshift variants affect the ability of hnRNPA2 to bind Kapβ2 and thus impair its subsequent nuclear import.

**Frameshift variants cause apoptotic cell death in differentiating cells.** To address the consequence of the frameshift mutants in a physiologically relevant cell type, we expressed WT or mutant versions of hnRNPA2 in C2C12 myoblasts and subsequently exposed the cells to differentiation factors to induce differentiation to myotubes (Fig. 4a). Whereas there was no obvious impact of exogenous expression of WT or mutant hnRNPA2 in myoblast cells, we observed that upon differentiation, the number of myotubes expressing mutant forms of hnRNPA2 decreased dramatically over time (Fig. 4b). Thus, to determine whether mutant forms of hnRNPA2 cause cell death in differentiating C2C12 cells, we monitored annexin V and propidium iodide (PI) staining to assess apoptotic cell death. Notably, cell death was significantly enhanced in the presence of the frameshift variants, with approximately 60% and 80% of frameshift mutant hnRNPA2-expressing cells becoming positive for annexin V and PI staining one and two days after differentiation, respectively (Fig. 4c, d, Supplementary Fig. 5a). In contrast, no increased cell death was observed in the overall population of cells expressing hnRNPA2 WT (Fig. 4c, d). However, when we specifically examined subpopulations of cells based on expression levels, we noted that the population of cells with the highest expression of hnRNPA2 WT did show increased cell death relative to controls (Fig. 4e). In cells expressing frameshift mutant forms of hnRNPA2, increased cell death was observed even in the population of cells with the lowest levels of mutant hnRNPA2 expression (Fig. 4e). We also examined the consequences of depleting endogenous hnRNPA2. Specifically, we found that depletion of wild-type endogenous hnRNPA2/B1 by expressing

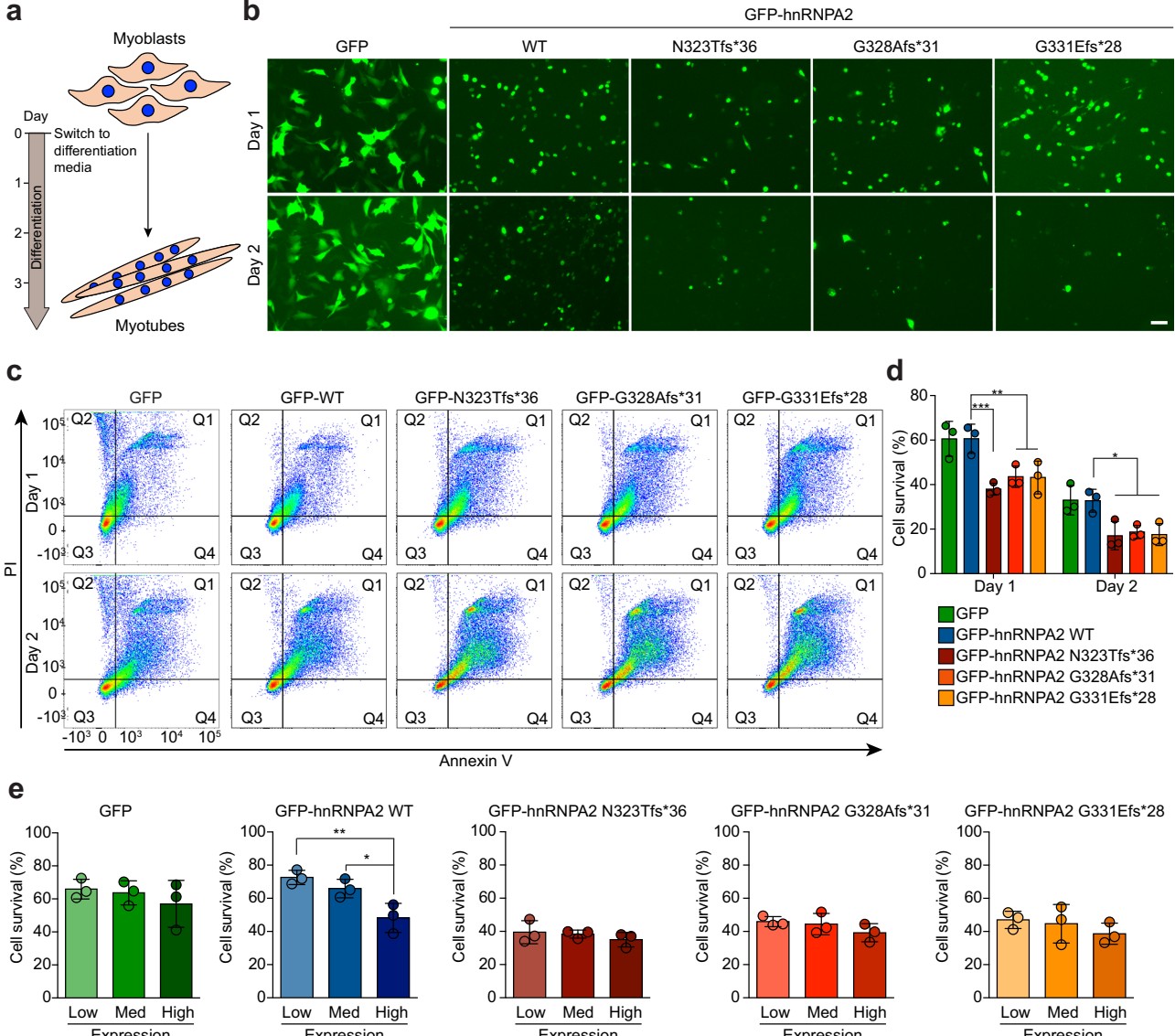

**Fig. 4 Expression of frameshift variants causes increased cell toxicity during C2C12 differentiation. a** Schematic of C2C12 differentiation. Differentiation media consists of DMEM supplemented with 0.5% FBS, 1% penicillin/streptomycin, and 1% L-glutamate. **b** Representative images of C2C12 cells one (top) or two (bottom) days after change to differentiation media from three independent experiments. Scale bar, 20 μm. **c** Representative scatter plots of GFP-positive C2C12 cells stained with propidium iodide (PI) and annexin V, one (top) or two (bottom) days after change to differentiation media. **d** Quantification of GFP-positive cells that were negative for both PI and annexin V. Values represent means ± s.e.m. ($n = 3$ independent experiments). ***$P = 0.0004$, **$P = 0.0061$, and **$P = 0.0052$ (day 1) and *$P = 0.0116$, *$P = 0.0252$, and *$P = 0.0147$ (day 2) for N323Tfs*36, G328Afs*31, and G331Efs*28 mutants, respectively, by two-way ANOVA with Dunnett's multiple comparisons test. **e** Quantification of cell survival for all GFP-positive cells from day 1 post differentiation split into three categories. The range of GFP intensity was 500–140,000. Cells whose GFP intensity was below 45,000, between 45,000 and 90,000, or above 90,000 were grouped into low, medium, or high GFP-expressing cells, respectively. Values represent means ± s.e.m. from $n = 3$ independent experiments. *$P = 0.0364$ and **$P = 0.0088$ by one-way ANOVA with Tukey's multiple comparisons test.

small interfering RNA against *HNRNPA2B1* did not cause cell toxicity (Supplementary Fig. 5b–d). These data suggest that despite the loss of Kapβ2 binding activity by frameshift mutations, the mechanism of toxicity associated with these mutations is mediated by a gain of function rather than simply a loss of function.

**No evidence that the frameshift mutations introduce a specific toxic sequence.** We found it intriguing that all of the eoOPMD-associated variants we identified had a +1 frameshift resulting in a neomorphic amino acid sequence with a shared C-terminal end (Fig. 3b). This alteration introduces several negatively

charged amino acids and makes the C-terminal peptide more hydrophobic compared to the WT peptide (Fig. 5a, b). It is conceivable, therefore, that the precise sequence introduced by the frameshift mutation is responsible for the toxic gain of function. Thus, we tested whether the anomalous sequence was directly responsible for the phenotypes associated with frameshift variants of hnRNPA2. To this end, we generated two new constructs in which the C-terminal flanking sequence was either deleted (Δ323–341) or +2 frameshifted by a two-base-pair deletion (N323Lfs*31) (Fig. 5b). Interestingly, both Δ323–341 and N323Lfs*31 proteins accumulated in the cytoplasm (Fig. 5c) and caused cell toxicity in differentiated myoblasts at levels comparable to the eoOPMD-associated frameshift variants

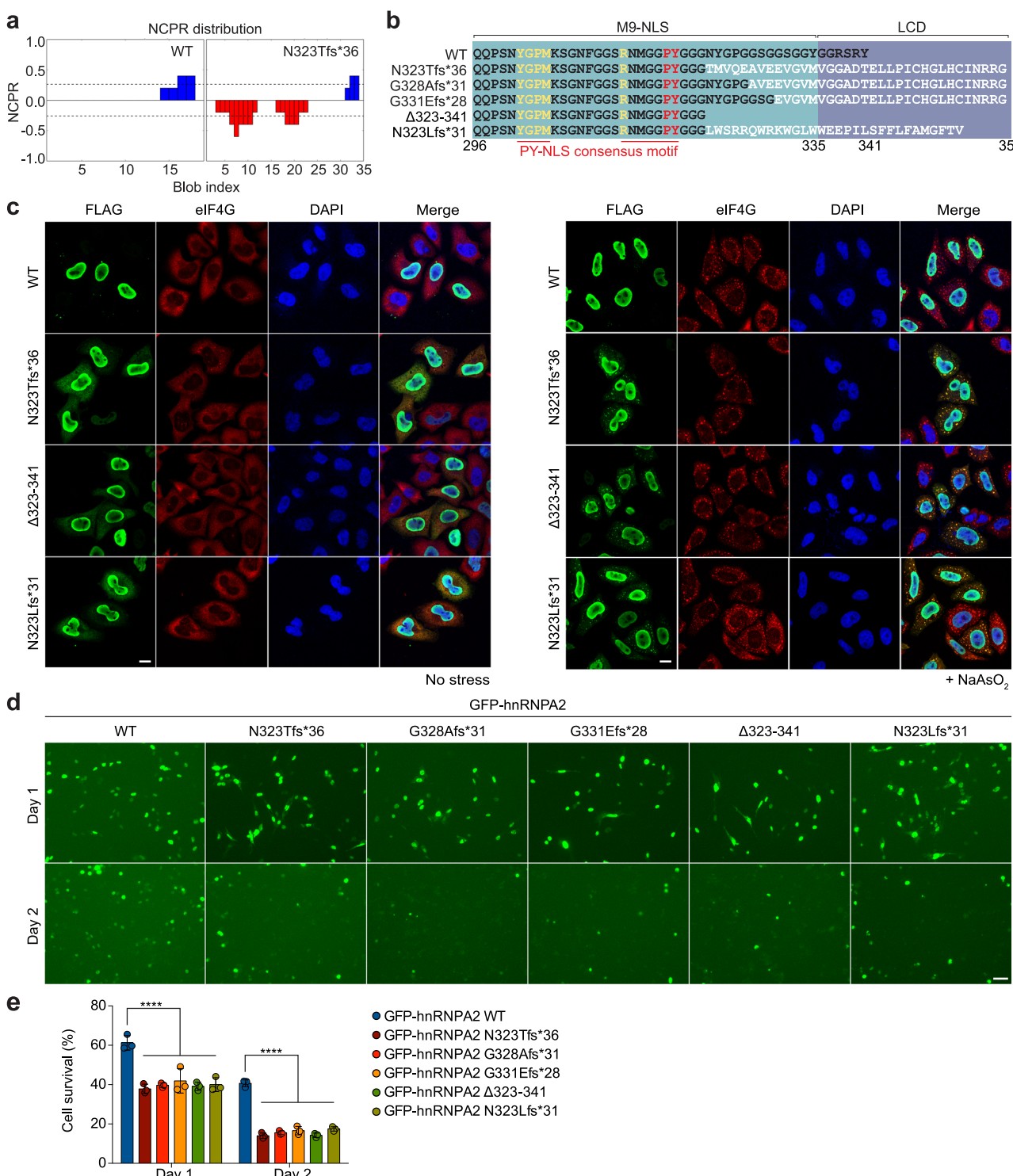

**Fig. 5 A loss of wild-type C-terminal sequence, not a neomorphic sequence, is responsible for frameshift variant phenotypes. a** NCPR (net charge per residue; 5-aa window) of WT and N323Tfs*36 peptides. Positively (blue) and negatively (red) charged amino acids are indicated. **b** Sequences in WT, +1 frameshift mutants in eoOPMD, Δ323–341, and +2 frameshift mutant (N323Lfs*31). **c** Localization of FLAG-tagged hnRNPA2 proteins without (left) or with (right) 0.5 mM NaAsO₂. eIF4G was used as a cytoplasmic and stress granule marker. Representative images from three independent experiments are shown. Scale bar, 10 μm. **d** Representative images of C2C12 cells one (top) or two (bottom) days after change to differentiation media from three independent experiments. Scale bar, 20 μm. **e** Quantification of GFP-positive cells negative for both PI and annexin V. Values represent means ± s.e.m. (*n* = 3 independent experiments). ****P < 0.0001 by two-way ANOVA with Dunnett's multiple comparisons test.

(Fig. 5d, e, Supplementary Fig. 5e). This result suggests that loss of the wild-type C-terminal flanking sequence, rather than addition of a specific, negatively charged C-terminal sequence, is responsible for the observed cellular phenotypes associated with *HNRNPA2B1* frameshift variants.

**Altered nucleocytoplasmic distribution of hnRNPA2 is responsible for the cytotoxicity associated with frameshift variants.** To further investigate the contribution of nucleocytoplasmic transport defects to the toxic mechanism of frameshift variants, we generated a series of additional PY-NLS mutants. These included the introduction of missense mutations (P318A and Y319A) to or deletions (ΔPY and ΔPY-NLS) of key residues in the NLS. We found that a single amino acid substitution (P318A or Y319A) or a deletion (ΔPY) of the PY motif did not alter nuclear localization of hnRNPA2 (Fig. 6a, b)[37]. In contrast, the ΔPY-NLS mutant (in which residues spanning the two conserved PY-NLS consensus motifs, **YGPM**KSGNFGGS**R**NMGG**PY** [aa 301–319], were deleted) showed cytoplasmic accumulation of the protein that was comparable to the N323Tfs*36 variant (Fig. 6a, b). The Δ323–341 mutant (in which the C-terminal flanking sequence was deleted) also showed cytoplasmic accumulation of the protein that was comparable to the N323Tfs*36 variant (Fig. 5b, c). Taken together, these results confirm the importance of PY-NLS flanking sequences in binding Kapβ2 and are consistent with previous structural studies of hnRNPA1 that revealed that in addition to modest energetic contributions of the PY residues to binding Kapβ2, important electrostatic interactions are also provided by the adjacent C-terminal flanking regions[34,35]. Importantly, the partial shift to cytoplasmic localization of hnRNPA2 was sufficient to drive toxicity. Indeed, expression of the artificial ΔPY-NLS mutant in differentiated C2C12 cells, which causes modest cytoplasmic accumulation of this protein, resulted in cellular toxicity similar to the frameshift mutants (Fig. 6c, d).

To complement these additional PY-NLS mutants, we also designed a rescue experiment. Specifically, we added the canonical monopartite cNLS sequence from simian virus 40 (SV40) T-antigen[38] to the C terminus of N323Tfs*36. Not only did appending the NLS sequence onto this mutant hnRNPA2 fully rescue nuclear localization of N323Tfs*36 (Fig. 6a, b), but it also prevented cell toxicity (Fig. 6c, d). Thus, we conclude that the wild-type C-terminal flanking sequence is important for nuclear localization of hnRNPA2 by regulating its interaction with the nuclear transport receptor Kapβ2, that this interaction is partly impaired by frameshift mutation, and that the altered distribution of hnRNPA2 leads to cell toxicity.

**Frameshift variants cause eye and muscle degeneration in a *Drosophila* model.** We next investigated the in vivo effects of missense and frameshift mutations in a *Drosophila* model system. To this end, we generated transgenic *Drosophila* expressing human hnRNPA2 WT, the MSP-associated mutant D290V[6], or frameshift mutants (N323Tfs*36, G328Afs*31, and G331Efs*28) via PhiC31 integrase-mediated site-specific insertion of a single copy of the human *HNRNPA2B1* gene. We established multiple fly lines per genotype and observed that all lines expressing the N323Tfs*36 mutant, and some lines expressing G328Afs*31 and G331Efs*28, showed low or no protein expression with the eye tissue-targeting GMR-GAL4 driver, which may reflect cytotoxicity associated with expression (Supplementary Fig. 6a). From the remaining lines, we selected one G328Afs*31 line and one G331Efs*28 line for subsequent analysis and compared the consequences of transgene expression with flies expressing

similar levels of hnRNPA2 WT or D290V (Supplementary Fig. 6a).

Expression of these transgenes in the fly eye revealed a mutation-dependent rough eye phenotype that was modest at 22 °C but enhanced at 25 °C, a temperature that causes a greater level of transgene expression (Fig. 7a)[39]. At 25 °C, flies expressing G328Afs*31 showed pupal lethality, although one lethality "escaper" G328Afs*31-expressing fly exhibited severe eye degeneration (Fig. 7a). When expression was driven by the muscle-specific driver MHC-GAL4, we observed robust expression of N323Tfs*36, G328Afs*31, and G331Efs*28 mutant proteins, in contrast with the low expression induced by GMR-GAL4 (Supplementary Fig. 6b). Although a single copy of the *HNRNPA2B1* gene was inserted in all lines, hnRNPA2 protein levels were consistently modestly higher in the N323Tfs*36, G328Afs*31, and G331Efs*28 flies compared with the hnRNPA2 WT and D290V flies, suggesting that the frameshift mutations may increase the stability of hnRNPA2 protein in fly muscles (Supplementary Fig. 6b).

A mutation-dependent phenotype was more obvious when the transgenes were expressed in skeletal muscle and assessed by wing position, as previously described[40]. Whereas flies expressing hnRNPA2 WT had normal wing position, flies expressing either hnRNPA2 with a missense mutation (D290V) or a frameshift mutation (N323Tfs*36, G328Afs*31, or G331Efs*28) exhibited a wing position defect. In these animals, a mild mutation-dependent wing position defect was evident on the first day after hatching (eclosion) and was found to affect up to 100% of the animals by day 5 after eclosion (Fig. 7b).

Consistent with the normal wing position, expression of hnRNPA2 WT in skeletal muscle caused no histological defect in adult skeletal muscle (Fig. 7c). Tissue architecture and myofibril organization were normal, and hnRNPA2 protein was detected only in the nucleus (Fig. 7c). In contrast, expression of either the missense or frameshift mutants caused a significant disruption in cytoarchitecture as evidenced by defects in the organization of myofibrils and substantial redistribution of the protein from the nucleus into the cytoplasm (Fig. 7c). However, there was a considerable difference in the cytoplasmic distribution of the hnRNPA2 protein in animals expressing missense vs. frameshift mutants. The missense mutant (D290V) accumulated in dense sarcoplasmic punctae, consistent with our prior observation in both *Drosophila* and human cells that hnRNPA2 D290V accumulates in sarcoplasmic RNP granules[6,41]. In contrast, the frameshift mutants accumulated in the sarcoplasm with a diffuse distribution (Fig. 7c–e).

We previously showed that the missense (D290V) mutant hnRNPA2 protein shows a stark reduction in solubility relative to WT when expressed in either human HeLa cells or *Drosophila* skeletal muscle cells[6]. In contrast, frameshift (N323Tfs*36, G328Afs*31, and G331Efs*28) mutants showed no reduction in solubility in *Drosophila* skeletal muscle cells (Fig. 7f, g) and indeed showed somewhat increased solubility when expressed in human cells (Supplementary Fig. 4e). These results suggest that the consequences of these two classes of mutations on intrinsic properties of hnRNPA2 protein may be very different.

To directly address the consequences of the different classes of mutation on the biophysical properties of hnRNPA2, we examined the fibrillization kinetics of purified proteins. These results revealed that frameshift variants exhibited decelerated fibrillization kinetics compared to WT protein, whereas hnRNPA2 D290V assembled into fibrils at a faster rate, as previously reported (Supplementary Fig. 7)[6]. Specifically, the N323Tfs*36 mutant formed very few fibrils over 24 h, whereas the G328Afs*31 mutant fibrillized only after a long lag phase

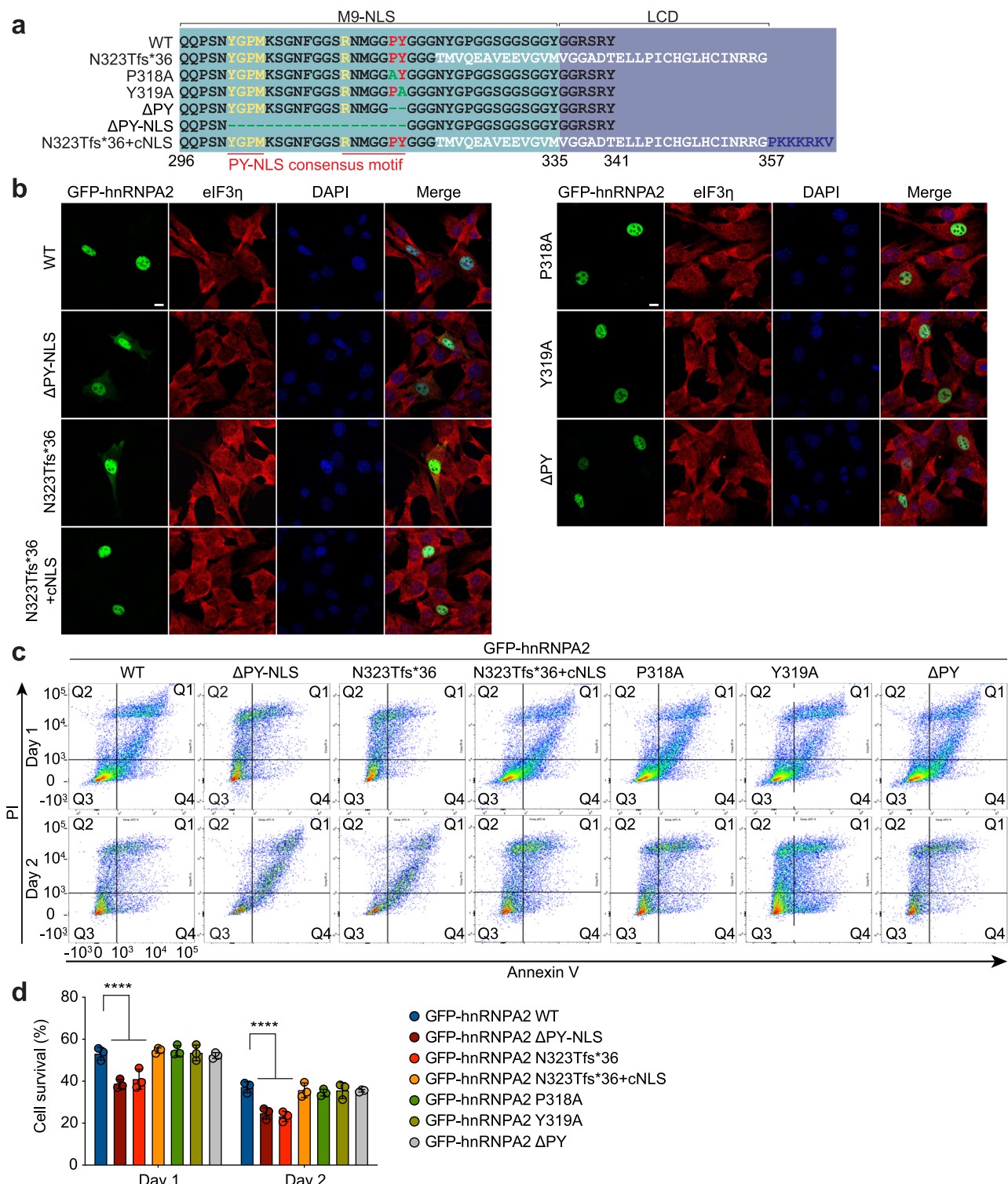

**Fig. 6 Altered nucleocytoplasmic distribution of hnRNPA2 is responsible for the cytotoxicity associated with its frameshift variants. a** The amino acid sequences of the relevant domains in WT, N323Tfs*36, P318A, Y319A, ΔPY, ΔPY-NLS and N323Tfs*36+cNLS. **b** Intracellular localization of FLAG-tagged hnRNPA2 proteins. Representative images from three independent experiments are shown. Scale bar, 10 μm. **c** Representative scatter plots of GFP-positive C2C12 cells stained with propidium iodide (PI) and annexin V, one (top) or two (bottom) days after change to differentiation media. **d** Quantification of GFP-positive cells that were negative for both PI and annexin V. Values represent means ± s.e.m. ($n = 3$ independent experiments). ****$P < 0.0001$ by two-way ANOVA with Dunnett's multiple comparisons test.

(Supplementary Fig. 7a, b). Thus, these findings suggest that the pathogenic mechanism of the frameshift variants differs from that of D290V and likely involves reduced efficiency of nuclear import rather than an increased propensity toward fibrillization.

## Discussion

Here we describe a distinct phenotype of early-onset myopathy caused by specific heterozygous *HNRNPA2B1* frameshift mutations clinically manifesting with progressive muscle weakness,

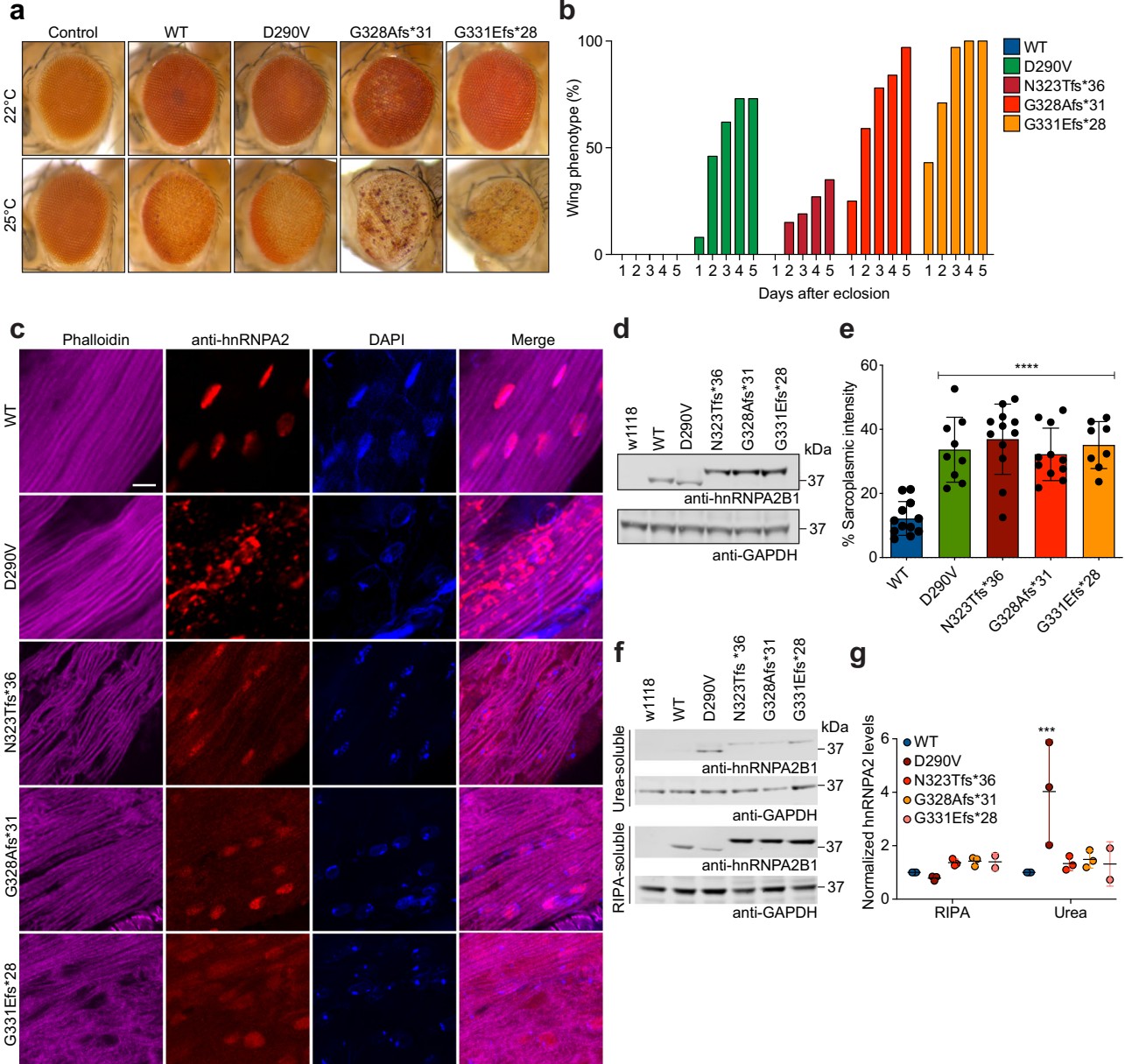

**Fig. 7 Frameshift variants cause eye and muscle degeneration in a *Drosophila* model. a** Expression of frameshift variants in *Drosophila* eye tissue using GMR-GAL4 causes a rough eye phenotype. **b** Expression of frameshift variants in *Drosophila* muscle using MHC-GAL4 causes an abnormal wing posture phenotype. The percentage of flies with abnormal wing posture is plotted for 5 days after eclosion. $n = 26, 26, 26, 32,$ and 35 flies for WT, D290V, N323Tfs*36, G328Afs*31, and G331Efs*28, respectively. **c** Adult flies expressing hnRNPA2 transgene using MHC-GAL4 were dissected to expose the dorsal longitudinal indirect flight muscle and stained with rhodamine-phalloidin (purple), hnRNPA2 (red), and DAPI (blue). hnRNPA2 WT localized exclusively to nuclei, whereas hnRNPA2 D290V accumulated extensively in cytoplasmic inclusions. Frameshift mutants showed both nuclear staining and diffuse sarcoplasmic accumulation. Scale bar, 10 μm. **d** hnRNPA2 expression in thoraces of adult flies driven by MHC-GAL4 in transgenic flies. Representative immunoblots from three independent experiments are shown. **e** Ratio of sarcoplasmic intensity of hnRNPA2 signal in indirect flight muscles. Error bars represent mean ± s.d. $n = 12, 9, 12, 11,$ and 8 fly muscle samples for WT, D290V, N323Tfs*36, G328Afs*31, and G331Efs*28, respectively. ****$P < 0.0001$ by one-way ANOVA with Dunnett's multiple comparisons test. **f** Thoraces of adult flies expressing hnRNPA2 transgene using MHC-GAL4 were dissected and sequential extractions were performed to examine the solubility profile of hnRNPA2. w1118 flies are included as a non-transgenic control. Representative immunoblots from three independent experiments are shown. **g** Quantification of RIPA-soluble and -insoluble fraction of hnRNPA2. Data represent mean ± s.d., $n = 3$ (WT, D290V, N323Tfs*36, G328Afs*31) and 2 (G331Efs*28) independent experiments. ***$P = 0.0002$ by two-way ANOVA with Dunnett's multiple comparisons test.

ptosis, ophthalmoplegia, dysphagia, and variable degrees of respiratory insufficiency. With the notable exception of the very early disease onset and rapid progression in our cohort, the overall clinical presentation was reminiscent of classical OPMD, such that the disease could be understood as a distinct early-onset

form of OPMD (eoOPMD). The uniform absence of bone, cognitive and motor neuron involvement in our patients distinguishes our cohort from the previously described MSP phenotype associated with a p.D290V *HNRNPA2B1* variant[6]. In addition, ptosis, ophthalmoparesis, and dysphagia were absent in the

reported *HNRNPA2B1* missense (D290V) family, whereas these were consistent phenotypic features within our patient cohort, further differentiating these distinct *HNRNPA2B1*-related phenotypes.

Muscle pathologic features in our *HNRNPA2B1* cohort were overall consistent with a chronic, degenerative myopathy, with notable histological findings of rimmed autophagic vacuoles and cytoplasmic and intranuclear tubulofilamentous inclusions. These non-specific findings have been reported in several myopathies, including *HNRNPA2B1* missense variant MSP, inclusion body myositis[42], MYH2-related inclusion body myopathy[43], hnRNPA1-associated inclusion body myopathy[44], TIA1-related myopathies[45], hnRNPDL-related myopathies[11], and OPMD[46], although the intranuclear tubulofilamentous inclusions we noted lacked OPMD-associated palisading morphology. The relevance of autophagic vacuoles and tubular filamentous inclusions to the pathophysiology of the specific myopathic phenotype observed in our patients remains unclear.

Since the identification of the first *HNRNPA2B1* MSP family, screening of MSP and large ALS patient cohorts has shown that *HNRNPA2B1* variants are a rare cause of sporadic and familial motor neuron disease[47–49]; however, its potential role in myopathies has remained unclear. Degenerative myopathy was observed in several family members carrying the missense D290V mutations[6]. More recently, a p.P298L missense *HNRNPA2B1* mutation was found to cause isolated familial PDB without multisystem involvement, highlighting the divergent phenotypes that may arise from the various *HNRNPA2B1* disease variants[10]. In contrast with previously reported variants, the *HNRNPA2B1* variants reported here are all specific frameshift mutations that tightly cluster in the highly conserved LCD of the protein and escape RNA quality control degradation (Supplementary Fig. 2). These frameshift *HNRNPA2B1* variants alter the C-terminal M9-NLS amino acid sequence of the protein, while sparing the immediately upstream PY residues. Further investigation of the sequence features that contribute to frameshift variant-associated phenotypes indicated that partial loss of Kapβ2 binding, rather than gain of a neomorphic amino acid sequence in frameshift variants, is responsible for the cellular phenotypes observed (Fig. 5, Supplementary Fig. 5). Thus, we suggest that the C-terminal flanking sequence after the PY-NLS of hnRNPA2/B1 contributes to Kapβ2 binding, similar to the molecular interaction of the closely related hnRNPA1 protein with Kapβ2[34,35]. We propose further that loss of this C-terminal flanking sequence is responsible for the partial loss of interaction with Kapβ2, which is the most proximal consequence of the mutations and initiates the slow pathological process. It remains to be seen whether the same eoOPMD phenotype is observed in rare individuals with pure truncations or alternative frameshifts in the *HNRNPA2B1* C-terminal flanking sequence. While it is possible that loss of hnRNPA2/B1 nuclear function may contribute to disease progression, the overall reduction in nuclear localization of hnRNPA2/B1 is modest. Moreover, we found that strong knockdown of endogenous hnRNPA2/B1 is well tolerated in differentiating myotubes, at least on a time scale of days. Rather, the data presented herein suggests that the disease mechanism promoted by frameshift mutations is toxic gain of function in the cytoplasm. It is important to note that Kapβ2 not only regulates nuclear import of hnRNPA2/B1 (and other clients such as hnRNPA1 and FUS) but also regulates higher-order assembly of these RBPs via LLPS and fibrillization[50–52]. Thus it is possible that loss of Kapβ2-mediated chaperoning of hnRNPA2/B1 in the cytoplasm may contribute to pathogenesis, and this will require further investigation.

The similarity in muscle pathological features of *HNRNPA2B1*-associated MSP and eoOPMD may appear paradoxical, as these

two classes of mutation have distinct consequences for the hnRNPA2 protein: the first (missense mutations in the LCD sequence) increases the propensity toward fibrillization, whereas the second (frameshift mutations that partially impair nuclear import) increases the cytoplasmic concentration. The long-term consequence of both classes of mutations is the accrual of amorphous aggregates that include the hnRNPA2 protein. Thus, our working model is that there are two distinct, but not mutually exclusive mechanisms that can lead to pathological proteinaceous deposits[20,21]. First, pathological aggregates can arise from mutations that reduce the energy threshold for fibrillization. In this category are mutations that alter key residues in the LCDs of RBPs (e.g., D290V in hnRNPA2/B1, A315T in TDP-43). Second, pathological aggregates can also arise through the evolution of poorly dynamic RNP granules. RNP granules are dynamic liquids that, when perturbed, can undergo a liquid-to-solid phase transition that culminates in pathological aggregates similar to those that arise through primary nucleation. Such perturbations include increased concentration of constituent proteins (e.g., RBPs) and impairment of regulatory factors (e.g., VCP). Disease-causing mutations in this second category include (1) NLS mutations that increase the cytosolic concentration of RBPs (e.g., the frameshift mutations in hnRNPA2/B1 reported here, and a series of mutations in FUS that are clustered at the NLS), (2) hexanucleotide expansion in *C9ORF72* that produces dipeptide repeats that insinuate into biomolecular condensates and alter their material properties[53,54], and (3) mutations in VCP that impair its ability to disassemble RNP granules[55]. Although the nature and direct molecular consequences of different classes of mutation are clearly different (e.g., D290V vs. the eoOPMD frameshift variants in *HNRNPA2B1*), mutations in both categories can ultimately lead to pathological phase transition and deposition of protein inclusions over time. Indeed, in human pathological studies of ALS/FTD brain samples, which at late stages are also characterized by presence of TDP-43 cytoplasmic aggregates, loss of nuclear TDP-43 staining precedes the development of cytoplasmic inclusions by several years[56]. Thus, we suggest that the hnRNPA2/B1 aggregates in our patient biopsies are the end-stage epiphenomenon of the disease process over a long period of time and are preceded by abnormal nucleocytoplasmic transport dynamics of frameshift variant hnRNPA2/B1 and its accumulation in the cytoplasm. This may also explain why a plethora of RBPs in RNP granules lead to co-deposition of additional RBPs (e.g., TIA1 and TDP-43) that appear as pathological features.

All the *HNRNPA2B1* frameshift variants reported here were found in heterozygosity and thus appear to act in a dominant manner. This observation raises the possibility of loss of function (i.e., haploinsufficiency), gain of function, or dominant-negative pathomechanisms. As noted above, missense mutations in the LCD of hnRNPA2 may drive primary nucleation of hnRNPA2 fibrils and culminate in the accrual of pathological aggregates, reflecting a straightforward gain of toxic function. In contrast, frameshift variants in hnRNPA2 partially impair nuclear import —a partial loss of function. However, the consequence of this impaired nuclear import is persistently increased concentration of hnRNPA2 in the cytoplasm. Substantial evidence has now accrued that increased concentration of an RNP granule constituent is one of several mechanisms that impairs the dynamics of RNP granules, culminating in liquid-to-solid phase transition and evolution toward pathological aggregates that also include hnRNPA2 protein[20,21]. Thus, whereas the primary consequence of the frameshift mutations is loss of function (impaired binding to Kapβ2), we suggest that the resulting increased cytoplasmic concentration ultimately results in the same consequences as the

missense mutations—a toxic gain of function in the cytoplasm. This model is further supported by the results of knockdown experiments showing that depletion of endogenous *HNRNPA2B1* by siRNA does not cause toxicity in differentiated C2C12 cells, arguing against a simple loss of function mechanism (Supplementary Fig. 5b–d).

It is noteworthy that similar C-terminal frameshift variants in *HNRNPA1* (*321Eext*6, *321Qext*6, and G304Nfs*3) were identified in a broad spectrum of patients with hereditary motor neuropathy, ALS and myopathy[7]. The amino acid sequences altered by these frameshift mutations in hnRNPA1 are located far downstream of the PY-NLS, and the impact of these frameshift mutations on nucleocytoplasmic transport of hnRNPA1 is not as clear as observed for the *HNRNPA2B1* frameshift variants. Nevertheless, similar to the eoOPMD-associated *HNRNPA2B1* frameshift variants, frameshift variants of *HNRNPA1* showed reduced propensity for fibrillization, and one variant with an extension of amino acids (*321Eext*6) accumulated in cytoplasmic stress granules in response to stress, which can often be attributed to increased concentration of the protein in the cytoplasm[7]. Thus, the frameshift variants of *HNRNPA1* and *HNRNPA2B1* might share a common underlying pathomechanism.

Among the myopathies, emerging data suggest a specific role for LCD-containing RBPs such as TDP-43 and hnRNPA2/B1 in regenerating muscle, where they are thought to stabilize large muscle-specific transcripts (e.g., TTN, NEB), aid in their transport, and facilitate pre-mRNA splicing[57]. In particular, wild-type hnRNPA2/B1 retains its exclusive nuclear localization during muscle regeneration[57]. In association with their altered nucleo-cytoplasmic distribution, expression of *HNRNPA2B1* frameshift variants in our study resulted in cell toxicity in differentiating C2C12 myoblasts (Fig. 4) and degeneration of flight muscle in *Drosophila* (Fig. 7), suggesting an essential role for hnRNPA2/B1 in maintaining skeletal muscle integrity. Thus, with ongoing use and stress of the muscle, impaired hnRNPA2/B1 nuclear import may be inadequate to maintain muscle homeostasis over time, clinically manifesting as the progressive muscle weakness seen in our patients. The phenotypic relationship of our patients to typical OPMD is also of great interest, as OPMD is caused by polyalanine expansions in PABPN1, whose normal splicing in the nucleus is partially regulated by hnRNPA2/B1[58]. Thus, future studies should focus on the role of RBPs such as hnRNPA2/B1 in specific cell types, specific subcellular compartments, and in response to different physiologic states to identify their respective contributions to pathogenic processes and subsequent clinical manifestations.

Our data expand the clinical spectrum of *HNRNPA2B1* variants from MSP to include a distinct early-onset OPMD-like phenotype. Understanding how these seemingly divergent phenotypes emerge from common molecular and cellular events will likely uncover fundamental insights into RBP function and regulation that will be applicable to a broad array of neurodegenerative diseases.

## Methods

**Patient recruitment and sample collection.** Patients were recruited through local neurology and genetics clinics and clinical information was obtained based on the local standard clinical care. DNA and tissues (e.g., muscle, skin biopsy) and medical records were obtained based on standard procedures. The authors affirm that human research participants and/or legal guardian have seen and read the material to be published and have provided informed consent for publication of the images in Fig. 1. Ethical approval was obtained from the NIH, National Institute of Neurological Disorders and Stroke (NINDS), Institutional Review Broad (Protocol 12-N-0095), National Center of Neurology and Psychiatry (Protocol A2019-123), University of Strasburg (Protocol DC-2012-1693), Cambridge South, UK Research

Ethics Committee (approval 13/EE/0325), Health Research Authority, NRES Committee East of England—Hatfield (REC 13/EE/0398; REC 06/Q0406/33) and National Research Ethics Service (NRES) Committee North East–Newcastle & North Tyneside 1 (reference 08/H0906/28).

**Exome sequencing.** Quartet exome sequencing in family 1 was performed through the NIH Intramural Sequencing Center (NISC) using the Illumina TruSeq Exome Enrichment Kit and Illumina HiSeq 2500 sequencing instruments. Variants were analyzed using seqr (Center for Mendelian Genomics, Broad Institute). Trio exome sequencing in family 2 was performed at GeneDX with exon targets isolated by capture using the Agilent SureSelect Human All Exon V4 (50 Mb) kit or the Clinical Research Exome (Agilent Technologies). The sequencing methodology and variant interpretation protocol has been previously described[59].

Patient 3 was sequenced as a singleton as part of the MYO-SEQ project[60]. Exome sequencing was performed by the Genomics Platform at the Broad Institute. Libraries were created with an Illumina exome capture (38 Mb target) and sequenced with a mean target coverage of >80x. Exome sequencing data were analyzed on seqr (https://seqr.broadinstitute.org/).

For family 4, exome sequencing of patients 4 and 5 was undertaken using Agilent SureSelect Human All Exon 50 Mb capture kit followed by sequencing on a 5500XL SOLiD sequencer. The affected mother and unaffected father were sequenced using Nextera Rapid Capture Expanded Exome for target selection followed by sequencing on an Illumina HiSeq2000. Genomic data were processed as previously described[61]. Confirmation of variants and segregation was performed by Sanger sequencing.

WES analysis in proband (P6) from family 5 was performed by deCODE genetics, Iceland. The alignment to the human reference genome (hg19) and the variant calling was done by deCODE genetics, Iceland. Data analysis was carried out using Clinical Sequence Miner platform, NextCODE Health.

Trio exome sequencing in family 6 was performed at the Centre National de Recherche en Génomique Humaine (Evry, France) using the Agilent SureSelect Human All Exon V4. Sequence analysis and variant interpretation was performed as described[62].

For family 7, the patient underwent standard clinical next generation sequencing panels that included LGMD and congenital myopathy genes. Due to the remarkably similar phenotype to the other patients in this cohort, direct Sanger sequencing of the terminal exon of *HNRNPA2B1* was pursued and identified the de novo variant.

For family 8, the patient's DNA was analyzed using the Illumina Trusight ONE Expanded kit on an Illumina NextSeq 550 sequencer as described[63]. Segregation analysis was performed by Sanger sequencing.

For family 9, genome sequencing in patient 10 and her parents was performed using the Illumina HiSeq X Ten platform and variant calling and interpretation were performed as previously described[64]. Confirmation of variants was performed by Sanger sequencing.

For family 10, exome sequencing in patient 11 was performed at Beijing Genomics Institute (BGI) using the DNBSEQ. Exome sequencing and variant calling were performed as previously described[65]. Confirmation of variants was performed by Sanger sequencing.

**Muscle histology, immunofluorescence, and confocal microscopy.** Clinical muscle biopsy slides were obtained and reviewed. These included hematoxylin and eosin, modified Gömöri trichrome, NADH, and other histochemical stains as well as electron microscopy images when available. For immunofluorescence and confocal microscopy, frozen muscle biopsy tissues were cryo-sectioned (10 μm), fixed (100% acetone, −20 °C for 10 min), and blocked and permeabilized in 5% normal goat serum (Sigma) with 0.5% Triton X in PBS for 1 h at room temperature. Primary antibody incubation was performed overnight at 4 °C as follows: hnRNPA2B1 (Santa Cruz, sc-32316; mouse, 1:200), TDP-43 (Proteintech, 10782-2-AP; rabbit, 1:200), ubiquitin (Stressgen, SPA-200; rabbit, 1:200), ubiquilin-2 (Abcam, Ab190283; mouse IgG1, 1:500), TIA1 (Abcam, Ab140595; rabbit, 1:100), P62/SQSTM1 (Santa Cruz, sc-28359; mouse IgG1, 1:250). Secondary antibody incubation was performed for 1 h at room temperature. Sections were then washed in PBS, the nuclei were stained with DAPI, and sections were mounted and cover-slipped. Z-stack images were obtained using a Leica TCS SP5 II confocal microscope.

**Muscle MRI.** Muscle MRI was performed using conventional T1-weighted spin echo and short tau inversion recovery (STIR) of the lower extremities on different scanners at different centers.

**Plasmid constructs.** cDNA containing frameshift mutations of hnRNPA2 were synthesized by GenScript. For mammalian expression, FLAG-tagged WT, D290V, N323Tfs*36, G328Afs*31, G331Efs*28, Δ323–341, and N323Lfs*31 were cloned into pCAGGS vector at *SacI* and *SbfI* sites. N-terminal GFP-tagged WT, D290V, N323Tfs*36, G328Afs*31, and G331Efs*28 were cloned into pEGFP-C1 vector at *BsrGI* and *XhoI* sites. N-terminal GFP-tagged Δ323–341, N323Lfs*31, P318A, Y319A, ΔPY, ΔPY-NLS, and N323Tfs+cNLS were subjected to site-directed

mutagenesis using Q5 Site Directed Mutagenesis Kit (E0554S; NEB). C-terminal GFP-tagged WT, D290V, N323Tfs*36, G328Afs*31, and G331Efs*28 were cloned into pEGFP-N1 vector at *XhoI* and *PstI* sites. For transgenic *Drosophila*, mutant hnRNPA2 cDNAs were subcloned into the pUASTattB vector using *EcoRI* and *XhoI*. For bacterial expression, codon optimized cDNA containing frameshift mutations in hnRNPA2 were synthesized and subcloned into the pGST-Duet vector using *BamHI* and *EcoRI* by GenScript. All clones were verified by restriction enzyme digestion and sequence analysis.

**Cell lines**. The following cell lines were purchased from the American Type Culture Collection: HEK293T (ATCC CRL-11268), HeLa (ATCC CCL-2), and C2C12 (ATCC CRL-1772). Cells were authenticated by short tandem repeat profiling. All cell lines were tested to be mycoplasma negative.

**Cell culture, transfection, and immunofluorescence**. HEK293T and HeLa cells were grown in Dulbecco's modified Eagle's medium (DMEM) supplemented with 10% fetal bovine serum (FBS), 1% penicillin/streptomycin, and 1% L-glutamate. Cells were transfected using FuGene 6 (Promega) according to the manufacturer's instructions. For immunofluorescence, HeLa cells were seeded on eight-well glass slides (Millipore) and transfected with appropriate FLAG-tagged or GFP-tagged hnRNPA2 constructs. 24 h post transfection, cells were stressed with 500 µM sodium arsenite (Sigma-Aldrich) for indicated times. Cells were then fixed with 4% paraformaldehyde (Electron Microscopy Sciences), permeabilized with 0.5% Triton X-100, and blocked in 3% bovine serum albumin. Primary antibodies used were mouse monoclonal anti-FLAG (M2, F1804; Sigma), rabbit polyclonal anti-eIF4G (H-300, sc-11373; Santa Cruz Biotechnology), and mouse monoclonal anti-hnRNPA2B1 (EF-67, sc-53531; Santa Cruz Biotechnology) antibodies. For visualization, the appropriate host-specific Alexa Fluor 488, 555 or 647 (Molecular Probes) secondary antibody was used. Slides were mounted using Prolong Gold Antifade Reagent with DAPI (Life Technologies). Images were captured using a Leica TCS SP8 STED 3X confocal microscope (Leica Biosystems) with a ×63 objective. Images were quantified using Cell Profiler (Broad Institute) for cytoplasmic accumulation. Briefly, images were subjected to segmentation and integrated fluorescent intensity was calculated for whole cell, nucleus, and cytoplasm. Cytoplasmic percent is simply cytoplasmic intensity divided by whole cell intensity. For fluorescent distribution across the cell, ImageJ (NIH) was used. A straight line was overlaid across the cell and then the fluorescent intensity was measured across the line using the built-in function.

**Western blot analysis**. Cell lysates were prepared by lysing cells in 1× lysis buffer (150 mM NaCl, 25 mM Tris–HCl pH 7.5, 1 mM EDTA, 5% glycerol, and 1% NP-40) with Complete Protease Inhibitor Cocktail (Clontech Laboratories). Samples were resolved by electrophoresis on NuPAGE Novex 4–12% Bis–Tris gels (Invitrogen). Primary antibodies used were mouse monoclonal anti-FLAG (M2, F1804; Sigma), rabbit polyclonal anti-FLAG (F7425; Sigma), rabbit polyclonal anti-GFP (2555S; Cell Signaling), rabbit-polyclonal anti-hnRNPA2B1 (HPA001666, Sigma), rabbit-polyclonal anti-hnRNPA2B1 (NBP2-56497; NOVUS Biologicals), and mouse monoclonal anti-GAPDH antibodies (6C5, sc-32233; Santa Cruz Biotechnology). Blots were subsequently incubated with IRDye fluorescence-labeled secondary antibodies (LI-COR) and protein bands were visualized using the Odyssey Fc system (LI-COR) and Image Studio (LI-COR). Quantification was performed using ImageJ (NIH).

**Cell toxicity**. C2C12 cells were grown in Dulbecco's modified Eagle's medium (DMEM) supplemented with 10% fetal bovine serum (FBS), 1% penicillin/streptomycin, and 1% L-glutamate. Cells were transfected using Lipofectamine 300 reagent (ThermoFisher Scientific) according to the manufacturer's instructions. C2C12 cells were counted using ADAM-CellT (NanoEntek Inc., Seoul, Korea), plated in six-well dishes (Corning), and transfected with appropriate GFP-tagged hnRNPA2 constructs. 24 h post transfection, the media was changed to differentiation media (DMEM supplemented with 0.5% FBS, 1% penicillin/streptomycin, and 1% L-glutamate) and counted as day 0. After 1 or 2 days, the GFP signal of the cells were imaged with an EVOS microscope. Apoptotic cells were determined by staining with annexin-V-APC and propidium iodide (BD Biosciences) and measured by flow cytometry. Data was analyzed using FlowJo_v10.6.1.

**siRNA knockdown**. C2C12 cells were grown in Dulbecco's modified Eagle's medium (DMEM) supplemented with 10% fetal bovine serum (FBS), 1% penicillin/streptomycin and 1% L-glutamate. Cells were transfected using RNAiMAX reagent (ThermoFisher Scientific; 13778075) according to the manufacturer's instructions. C2C12 cells were counted using ADAM-CellT (NanoEntek Inc., Seoul, Korea), plated in six-well dishes (Corning), and transfected with either ON-TARGETplus Non-Targeting Pool siRNA (Dharmacon; D-001810-10-05) or ON-TARGETplus Mouse Hnrnpa2b1 siRNA (Dharmacon; L-040194-01-0005). 72 h post transfection, cells were used for subsequent experiments. Knockdown efficiency was determined by Western blot using mouse monoclonal anti-hnRNPA2/B1 (DP3B3, Santa Cruz Biotechnology; sc-32316) and goat polyclonal anti-actin (Santa Cruz Biotechnology; sc-1616). Target sequences for control siRNA were: UGGUUUAC AUGUCGACUAA, UGGUUUACAUGUUGUGUGA, UGGUUUACAUGUUU

UCUGA, UGGUUUACAUGUUUUCCUA. Target sequences for *HNRNPA2B1* siRNA were: GGAUCUGAUGGAUACGGAA, GGGAUGGCUAUAAUGGGUA, ACCGAUAGGCAGUCUGGAA, GGUGGAAUUAAGGAAGAUA.

**Protein purification**. GST-tagged Kapβ2 WT protein was purified as described previously[66] with modifications. Briefly, recombinant protein was expressed in BL21 (DE3) *Escherichia coli* cells by induction with 1 mM isopropyl-β-D-thiogalactoside overnight at 16 °C. Cells were lysed by sonication in buffer containing 50 mM HEPES pH 7.5, 150 mM NaCl, 2 mM EDTA, 2 mM DTT, 15% (v/v) glycerol with protease inhibitors and centrifuged. GST-Kapβ2 was then purified using Glutathione Sepharose 4B protein purification resin and eluted in buffer described previously with 30 mM glutathione, adjusted to pH 7.5. GST was cleaved using TEV protease at 4 °C or kept on the protein and Kapβ2 was further purified by ion-exchange and size exclusion chromatography in buffer containing 20 mM HEPES pH 7.5, 110 mM potassium acetate, 2 mM magnesium acetate, 2 mM DTT, and 10% (v/v) glycerol. Purified proteins were flash-frozen and stored at −80 °C. GST-tagged hnRNPA2 PY-NLS WT and N323Tfs*36 peptides were purified similarly except no ion-exchange step was included.

GST-tagged hnRNPA2 proteins were purified as described[6]. Briefly, WT and mutant hnRNPA2 with N-terminal GST tag were over-expressed into *E. coli* BL21(DE3)-RIL (Invitrogen). Bacteria were grown at 37 °C until reaching an OD$_{600}$ of ~0.6 and expression was induced by addition of 1 mM isopropyl 1-thio-β-D-galactopyranoside (IPTG) for 15–18 h at 15 °C. Protein was purified over a Glutathione-Sepharose column (GE) according to manufacturer instructions. GST-hnRNPA2 was eluted from the Glutathione Sepharose with 40 mM HEPES–NaOH, pH 7.4, 150 mM potassium chloride, 5% glycerol, and 20 mM reduced glutathione. Eluted proteins were centrifuged at 16,100×g for 10 min at 4 °C to remove any aggregated material before being flash-frozen in liquid N$_2$ and stored at −80 °C. Before each experiment, protein was centrifuged at 16,100×g for 10 min at 4 °C to remove any aggregated material. After centrifugation, the protein concentration in the supernatant was determined by Bradford assay (Bio-Rad) and these proteins were used for aggregation reactions. TEV protease was purified as described[67]. Briefly, His-tagged TEV were over-expressed into *E. coli* BL21(DE3)-RIL (Invitrogen). Bacteria were grown at 37 °C until reaching an OD$_{600}$ of ~0.7 and expression was induced by addition of 1 mM IPTG for 15–18 h at 15 °C. Protein was lysed in TEV Lysis Buffer (25 mM Tris–HCl, pH 8.0, 500 mM NaCl, 25 mM imidazole, 10 mM β-mercaptoethanol, with protease inhibitors) and purified over Ni-NTA resin (Qiagen). His-tagged TEV protein were eluted in buffer containing 25 mM Tris–HCl, pH 8.0, 500 mM NaCl, 300 mM imidazole, 10 mM β-mercaptoethanol. Protein were dialyzed into buffer containing 25 mM HEPES–NaOH, pH 7.0, 5 mM β-mercaptoethanol, 5% (v/v) glycerol, flash frozen, and stored at −80 °C.

**Fluorescence polarization**. Synthesized TAMRA-tagged hnRNPA2 PY-NLS WT, N323Tfs, G328Afs, or G331Efs peptide were incubated with increasing amounts of purified Kapβ2 WT in buffer containing 20 mM HEPES pH 7.5, 110 mM potassium acetate, 2 mM magnesium acetate, 2 mM DTT, and 10% (v/v) glycerol in Corning 96-well solid black polystyrene plates. Fluorescence polarization was measured with a Cytation 5 multi-mode plate reader (Biotek) using Gen5 software with 561-nm polarization cube. Analysis was performed using MATLAB. Briefly, fluorescence polarization was converted to anisotropy and $K_d$ values were calculated by fitting the resulting curve to the equation

$$F_{sb} = \frac{K_d + L + R - \sqrt{(K_d + L + R)^2 - 4 * L * R}}{2 * L};$$

where $F_{sb}$ is the fraction bound, $L$ is the concentration of peptide, and $R$ is the concentration of Kapβ2. Each peptide was run in triplicate.

**Generation of *Drosophila* lines and *Drosophila* stocks**. The hnRNPA2 WT and D290V *Drosophila* stocks have been previously published[6]. Flies carrying pUASTattB-hnRNPA2 N323Tfs, G328Afs, or G331Efs transgenes were generated by a standard injection and φC31 integrase-mediated transgenesis technique (BestGene Inc.). GMR-GAL4 was used to express transgenes in eyes; MHC-GAL4 was used to express transgenes in muscle. All *Drosophila* stocks were maintained in a 25 °C incubator with a 12-h day/night cycle.

**Adult *Drosophila* muscle preparation and immunohistochemistry**. Adult flies were embedded in a drop of OCT compound (Sakura Finetek) on a slide glass, frozen with liquid nitrogen, and bisected sagitally by a razor blade. After fixing with 4% paraformaldehyde in PBS, hemithoraces were stained by Texas Red-X phalloidin (Invitrogen) and DAPI according to manufacturer's instructions. Stained hemi-thoraces were mounted in 80% glycerol and the musculature was examined by DMIRE2 (Leica). For hnRNPA2 staining, hemithoraces were permeabilized with PBS containing 0.2% Triton X-100 and stained with anti-hnRNPA2B1 antibody (EF-67, sc-53531; Santa Cruz Biotechnology) and Alexa-488-conjugated secondary antibody (Invitrogen). Stained muscle fibers were dissected and mounted in Fluormount-G (Southern Biotech) and imaged with a Marianas confocal microscope (Zeiss).

**Solubility and biochemical analyses of adult *Drosophila* muscles and C-terminally GFP-tagged hnRNPA2 in C2C12 cells.** Sequential extractions were performed to examine the solubility profile of hnRNPA2. Adult fly thoraces or cells transfected with the appropriate GFP-tagged constructs were lysed in cold RIPA buffer (50 mM Tris, pH 7.5, 150 mM NaCl, 1% Triton X-100, 0.5% sodium deoxycholate, 0.1% SDS, and 1 mM EDTA) and sonicated. Cell lysates were cleared by centrifugation at 100,000×g for 30 min at 4 °C to generate RIPA-soluble samples. To prevent carry-overs, the resulting pellets were washed twice with PBS and RIPA-insoluble pellets were then extracted with urea buffer (7 M urea, 2 M thiourea, 4% CHAPS (3-[(3-cholamidopropyl)-dimethylammonio]-1-propanesulphonate), 30 mM Tris, pH 8.5), sonicated, and centrifuged at 100,000×g for 30 min at 22 °C. Protease inhibitors were added to all buffers before use. Protein concentration was determined by the bicinchoninic acid method (Pierce) and proteins were resolved by NuPAGE Novex 4–12% Bis–Tris gels (Invitrogen). For Western blot analysis, thoraces of adult flies were prepared and ground in NuPAGE LDS sample buffer (NP0007, Invitrogen). Samples were then boiled for 5 min and analyzed by standard Western blotting methods provided by Odyssey system (LI-COR) with 4–12% NuPAGE Bis–Tris gels (Invitrogen).

**In vitro fibrillization assays**. hnRNPA2 (5 μM) fibrillization (100 μl reaction) was initiated by addition of 1 μl of 2 mg/ml TEV protease in A2 assembly buffer (40 mM HEPES–NaOH, pH 7.4, 150 mM KCl, 5% glycerol, 1 mM DTT, and 20 mM glutathione). The hnRNPA2 fibrillization reactions were incubated at 25 °C for 0–24 h with agitation at 1200 rpm in an Eppendorf Thermomixer. For sedimentation analysis, at indicated time points, fibrillization reactions were centrifuged at 16,100×g for 10 min at 4 °C. Supernatant and pellet fractions were then resolved by SDS–PAGE and stained with Coomassie Brilliant Blue, and the relative amount in each fraction was determined by densitometry in ImageJ (NIH). All fibrillization assays were performed in triplicate. For electron microscopy, fibrillization reactions (10 μl) were adsorbed onto glow-discharged 300-mesh Formvar/carbon coated copper grids (Electron Microscopy Sciences) and stained with 2% (w/v) aqueous uranyl acetate. Excess liquid was removed and grids were allowed to air dry. Samples were viewed on a JEOL 1010 transmission electron microscope.

*Data collection and statistical analysis*. The NCPR was collected using CIDER v1.7 (http://pappulab.wustl.edu/CIDER/). Exome sequencing data were analyzed on seqr (https://seqr.broadinstitute.org/). FACS data was collected using BD FACS-Canto II Flow Cytometer (Version 9.0). All flow cytometry data were analyzed by FlowJo (Version 10.6.1). Images were quantified using Cell Profiler (Broad Institute, Version 4.1.3) and Image J (NIH, Version 2.1.0). Fluorescence polarization analysis was performed using MATLAB (Version 8.4). Statistical analyses were performed using Prism 9 (GraphPad, Version 9.3.1) software. Statistical tests used for individual experiments are described in corresponding legends.

**Reporting summary**. Further information on research design is available in the Nature Research Reporting Summary linked to this article.

## Data availability

Patient-related data, including genetic sequencing data not included in the manuscript or its supplements, were generated as part of clinical care and may be subject to patient confidentiality. All requests for raw and analyzed data and materials related to patients presented in this article will be reviewed by the respective institution to verify if the request is subject to any intellectual property or confidentiality obligations. Data requests for anonymized data are typically shared with qualified investigators after a material transfer agreement; such requests should be directed to corresponding author Carsten G. Bönnemann. All other data generated or analyzed during this study are included in this published article (and its Supplementary Information/Source Data file).

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

## Acknowledgements

We thank the patients and their families for their participation. We thank Christopher Mendoza and Gilberto ("Mike") Averion for their help in clinic. We also thank the NIH Intramural Sequencing Center staff and Daniel MacArthur and Fengmei Zhao (Analytic and Translational Genetics Unit at Massachusetts General Hospital in collaboration with the Broad Institute) for help with exome analysis. We thank Darren Chambers and Lucy Feng for technical assistance in processing muscle pathology specimens. We thank Natalia Nedelsky for editorial assistance. Work in C.G.B.'s laboratory is supported by intramural funds from the NIH National Institute of Neurological Disorders and Stroke. Exome sequencing was in part funded through the Clinical Center Genomics Opportunity, which is sponsored by the National Human Genome Research Institute, the NIH Deputy Director for Intramural Research, and the NIH Clinical Center. J.P.T. is supported by the Howard Hughes Medical Institute, R35NS097974, and the ALS Association (18-IIA-419). This study was supported in part by an Intramural Research Grant (2-5 and 29-4 to I.N.; 2-5 and 30-9 to A.I.) for Neurological and Psychiatric Disorders of NCNP and by AMED under grants 20ek0109490h0001 and JP19ek0109285h0003 (to I.N.) and Joint Usage and Joint Research Programs, the Institute of Advanced Medical Sciences, Tokushima University (2020, 2A19 to A.I.). Work in F.M.'s group was supported by the National Institute for Health Research Biomedical Research Centre at Great Ormond Street Hospital for Children NHS Foundation Trust and University College London and the MRC Centre for Neuromuscular Diseases Biobank, and by the HSS England Diagnostic and Advisory Service for Congenital Myopathies and Congenital Muscular Dystrophies in London, UK for their financial support to the DNC Muscle Pathology Service. F.M. was also funded by the European Community's Seventh Framework Program (FP7/2007–2013) under grant agreement 2012-305121 "Integrated European–omics research project for diagnosis and therapy in rare neuromuscular and neurodegenerative diseases (NEUROMICS)". E.P. and L.B. are members of the European Reference Network for Neuromuscular Diseases— Project ID 870177 and acknowledge support from the Telethon Network of Genetic BioBank (GTB12001D) and EuroBioBank network. The work in J.L.'s group is supported by the France Génomique National infrastructure and by the Fondation Maladies Rares within the frame of the "Myocapture" sequencing project, and by Association Française contre les Myopathies (22734). C.E.F. and F.L.R. are funded by Cambridge NIHR Biomedical Research Centre and the Rosetree Foundation. L.G. was supported by an Ellison Medical Foundation/American Federation for Aging Research fellowship, Alzheimer's Association Research fellowship, and a Target ALS Springboard Fellowship. C.M.F. was supported by NIH grants T32GM008275 and F31NS111870. A.F.F. was supported by NIH grants T32AG00255 and F31NS087676. J.S. was supported by Target ALS, Packard Foundation for ALS Research, The ALS Association, The G. Harold and Leila Y. Mathers Charitable Foundation, and NIH grant R01GM099836. MYOSEQ was funded by Sanofi Genzyme, Ultragenyx, LGMD2I Research Fund, Samantha J. Brazzo Foundation, LGMD2D Foundation and Kurt+Peter Foundation, Muscular Dystrophy UK, and Coalition to Cure Calpain 3. Analysis was provided by the Broad Institute Center for Mendelian Genomics and was funded by the National Human Genome Research Institute, the National Eye Institute, and the National Heart, Lung, and Blood Institute grant UM1 HG008900, and in part by National Human Genome Research Institute grant R01HG009141. The content is solely the responsibility of the authors and does not necessarily represent the official views of the National Institutes of Health.

## Author contributions

H.J.K., P.M., and S.D. designed and performed laboratory experiments, evaluated patients, analyzed data, and drafted the manuscript. K.O., L.G. and C.M.F. performed laboratory experiments, analyzed data, and drafted the manuscript. X.L., N.F., S.R.H., A.R.F., M.O., A.S., A.I., P.M., G.A., R.P., D.G.O., N.D., R.B., M.G. A.T., I.T.Z., L.B., T.E.L., A.K., M.S., A.K., S.M., P.Ma., Y.P., E.F., A.M., S.E., R.U.G., L.B., C.B., E.P., L.S., C.E.F., F.L.R., T.H., S.Q.-R., J.B., I.N., T.S., T.E., V.S., N.B.R., J.L., F.M., I.N., and M.A.T. provided and interpreted clinical and genetic information, crucial samples and material, and/or analyzed data. A.F.F., Y.H., and M.C. performed laboratory experiments, and/or analyzed data. J.S., C.G.B., and J.P.T. supervised the overall study, interpreted data, drafted, and revised the manuscript. All authors reviewed and edited the manuscript.

## Competing interests

J.P.T. is a consultant for 5AM and Third Rock Ventures. J.S. is a consultant for Dewpoint Therapeutics, ADRx, Neumora, Vivid Science, and Korro Bio. M.A.T. is the founder and CEO of Exerkine and CSO for Cora Therapeutics. The remaining authors declare no competing interests.

## Additional information

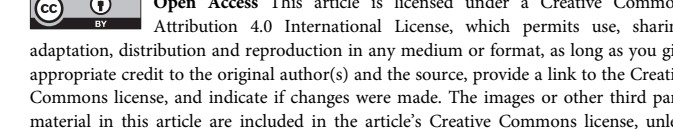

[1]Department of Cell and Molecular Biology, St. Jude Children's Research Hospital, Memphis, TN, United States. [2]National Institute of Neurological Disorders and Stroke, National Institutes of Health, Bethesda, MD, United States. [3]Department of Biochemistry & Biophysics, Perelman School of Medicine at the University of Pennsylvania, Philadelphia, PA, United States. [4]Department of Biochemistry and Molecular Biology, Thomas Jefferson University, Philadelphia, PA, United States. [5]Département Médecine Translationnelle et Neurogénétique, Institut de Génétique et de Biologie Moléculaire et Cellulaire, Institut National de la Santé et de la Recherche Médicale U1258, Centre National de la Recherche Scientifique UMR7104, Université de Strasbourg, Illkirch, France. [6]Wessex Clinical Genetics Services, Princess Anne Hospital, Academic Unit of Human Development and Health, Faculty of Medicine, University of Southampton, Southampton, England. [7]Wessex Neurological Centre, University Hospital Southampton, Southampton, UK. [8]Department of Neuromuscular Research, National Institute of Neuroscience, National Center of Neurology and Psychiatry (NCNP), 4-1-1 Ogawahigashi, Kodaira, Tokyo 187-8502, Japan. [9]Medical Genome Center, NCNP, Kodaira, Tokyo, Japan . [10]Department of Neurology, Niigata City General Hospital, Niigata, Japan. [11]The Dubowitz Neuromuscular Centre, NIHR Great Ormond Street Hospital Biomedical Research Centre, Great Ormond Street Institute of Child Health, University College London, & Great Ormond Street Hospital Trust, London, UK. [12]Department of Paediatric Neurology, Cambridge University Hospital NHS Trust, Addenbrookes Hospital, Cambridge CB2 0QQ, UK. [13]Division of Neuropathology, University College London Hospitals NHS Foundation Trust National Hospital for Neurology and Neurosurgery London, UK and Division of Neuropathology, UCL Institute of Neurology, Dubowitz Neuromuscular Centre, London, UK. [14]Department of Histopathology Box 235, Level 5 John Bonnett Clinical Laboratories Addenbrooke's Hospital, Cambridge, UK. [15]Institute of Medical Genetics and Applied Genomics, University of Tuebingen, Tuebingen, Germany. [16]John Walton Muscular Dystrophy Research Centre, Newcastle University and Newcastle Hospitals NHS Foundation Trust, Newcastle upon Tyne, UK. [17]Division of Neuromuscular & Neurometabolic Disorders, Department of Pediatrics, McMaster University, Hamilton Health Sciences Centre, Hamilton, ON, Canada. [18]Department of Neurology, Johns Hopkins University School of Medicine, Baltimore, MD, United States. [19]Division of Neuropaediatrics, Development and Rehabilitation, Department of Pediatrics, Inselspital, Bern University Hospital, University of Bern, Bern, Switzerland. [20]Pediatric Neurology, University Children's Hospital Basel, University of Basel, Basel, Switzerland. [21]Department of Neurometabolism, University Hospital of Nantes, Nantes, France. [22]CHU Nantes, Service de génétique médicale, Centre de Référence des Maladies Neuromusculaires AOC, 44000 Nantes, France. [23]Université de Nantes, CNRS, INSERM, l'institut du thorax, 44000 Nantes, France. [24]Service d'anatomopathologie, CHU Brest and EA 4685 LIEN, Université de Bretagne Occidentale, Brest, France. [25]CHU de Nantes, Centre de Référence des Maladies Neuromusculaires, Filnemus, Euro-NMD, Hôtel-Dieu, Nantes, France. [26]Etablissement de Santé pour Enfants et Adolescents de la région Nantaise, Nantes, France. [27]Human Genetics and Genomic Medicine, Faculty of Medicine, University of Southampton, Southampton, UK. [28]Department of Neurosciences, DNS, University of Padova, Padova, Italy. [29]Clinical Genetics Unit, Department of Women and Children's Health, University of Padova, IRP Città della Speranza, Padova, Italy. [30]Clinical Genetics Unit, Department of Women and Children's Health, CIR-Myo Myology Center, University of Padova, IRP Città della Speranza, Padova, Italy. [31]Department of Paediatrics, University of Cambridge, Cambridge, UK. [32]Institute of Neurology, Psychiatry and Narcology of NAMS of Ukraine, Kharkiv, Ukraine. [33]Cambridge Institute of Medical Research, University of Cambridge, Cambridge, UK. [34]Neuromuscular Unit, Pediatric Neurology and ICU Department, Raymond Poincaré Hospital (UVSQ), AP-HP Université Paris-Saclay, Garches, France. [35]Sorbonne Université, INSERM, Centre of Research in Myology, UMRS974 Paris, France. [36]APHP, Centre de Référence des Maladies Neuromusculaires Nord/Est/Ile de France, Institut de Myologie, Sorbonne Université, Hôpital Pitié-Salpêtrière, Paris, France. [37]Unité de Morphologie Neuromusculaire, Institut de Myologie, Sorbonne Université, Hôpital Pitié-Salpêtrière, Paris, France. [38]Howard Hughes Medical Institute, Chevy Chase, MD, United States. [39]These authors contributed equally: Hong Joo Kim, Payam Mohassel.
✉email: carsten.bonnemann@nih.gov; jpaul.taylor@stjude.org

