## [Peer Review File · Nature Communications]

Title: Heterozygous frameshift variants in HNRNPA2B1 cause early-onset oculopharyngeal muscular dystrophyREVIEWER COMMENTS

Reviewer #1 (Remarks to the Author):

Kim et al have manifestly proved their case that specific heterozygous frameshift variants close to the C-terminus of the protein in hnRNPA2B1 cause a characteristic myopathy that they have classified as early-onset oculopharyngeal muscular dystrophy. They describe 11 patients in ten families with the phenotype. All have the frameshift variants. In seven of the families, they demonstrate the variants arose de novo. Their functional genomic analyses indicate the pathomechanism of the frameshift variants is different to the pathomechanism of a missense-variants in the gene associated with Paget's disease of bone or multisystem proteinopathy.

There is little room to criticise the manuscript.

1) One change I would want to see is to include in the Summary an explanation of why the frameshift variants escape nonsense-mediated decay/RNA quality control degradation. This is not included at present. This leaves the reader desperate to know the reason, but they have to read on into the manuscript to obtain the answer.

Please include in the summary that the frameshift variants escape RNA quality control degradation, abolish the stop codon and extend the reading frame.

2) Whether the authors can say that the variants cause the same C-terminal rearrangement could be debated.

3) The authors should also discuss the number of exons in hnRNPA2B1 and which exon(s) the variants are in. For example, are the variants all in the last exon of the gene? Currently there is no discussion in the body of the manuscript of which exon the variants are in. In Supplementary figure 2, there is a diagram showing the variants are in exon 10, which appears to be the last coding exon, with the boundary between exon 10 and exon 11 being immediately after the stop codon. None of the relevance of this to the pathomechanism is discussed in the manuscript, but it should be.

4) It seems remarkable that the frameshift variants spaced so closely together in the gene can be classified according to ACMG guidelines as being pathogenic, likely pathogenic or of uncertain significance (Supplementary Table 1). The reasons for this should be discussed.

5) On page 14 of the PDF, the authors state: "the levels of protein expression were consistently modestly higher in the N323Tfs*36, G328Afs*31, and G331Efs*28 flies compared with the hnRNPA2 WT and D290V flies, suggesting that the frameshift mutations may increase the stability of hnRNPA2 protein in fly muscles". The explanation given as to why there is increased protein is that the frameshifted protein may be more stable. The explanation is not that the expression is higher. Therefore, please simply state: "the levels of protein were consistently modestly higher .."

6) The DAPI stain in Figure 6c is odd, it does not encompass the full extent of the anti-hnRNPA2, suggesting that the full extent of the nuclei has not been shown by the DAPI stain. Comment?

Reviewer #2 (Remarks to the Author):

The authors present exciting new results on frameshift variants of hnRNPA2B1 that are here shown to cause oculopharyngeal muscular dystrophy. The case for causation is clear with both genetic linkage and multiple similar variants in families/patients all disrupting the sequence immediately following the PY in the PY NLS of hnRNPA2. These variants are an interesting contrast to the missense variants found in hnRNPA2B1 and hnRNPA1 that cause MSP, suggesting that cytoplasmic localization may be more important than cytoplasmic aggregation to toxicity in cells. Overall the manuscript is exciting. The authors can consider the following questions that could improve the overall findings.

More insight in the biophysical mechanism. Does it really prevent aggregation in cells – more clear images showing D290V vs these frameshifts? I think the data are there, but it is somewhat unclear.

More insight in the toxicity mechanism. Is the PY-NLS still functional partially, is a delta PY-NLS toxic more or less, or is it something particular about deletion after the PY that leads to toxicity? What smaller deletions could be made to localize the residues that lead to disruption in binding and increased toxicity? It is somewhat surprising that the region after the PY is so important as several proteins have the final residues as PY (e.g. there is no flanking sequence).

The cell biology data with IF using two different antibodies (custom polyclonal surely fine) that distinguish the C-terminal regions of the proteins. It would help reveal the mechanism to understand if the mutants end up aggregating in the cell lines as well as in patient tissue.

Do the authors think similar variants will emerge in hnRNPA1?

The claim that fibrilization is slowed in vitro is clear in this experiment but it is not clear if this is a robust finding or if it is dependent on particular in vitro conditions that aren't applicable in cells. I suggest mentioning this possibility at least and saying these data "support" but maybe not "strongly suggestive" as the frameshifts also aggregate and in the TEM do not look that qualitatively different from WT.

"Cytoplasmic aggregate formation was associated with the solubility of hnRNPA2" This statement is not clear

Reviewer #3 (Remarks to the Author):

The authors describe a considerable cohort of 10 unrelated families with patients sharing a relatively distinct phenotype consisting of early onset progressive muscular dystrophy with ptosis, ophthalmoparesis and bulbar weakness, caused by different heterozygous frameshift mutations in the PrLd of the MSP gene HNRNPA1. This discovery defines a novel disease phenotype associated with the gene and in contrast with previous mutants these frameshifts seem to reduce the fibrillization properties of the mutants.

Concerns: There are no major concerns with this excellent work

Comments:

- Muscle MRI findings are described as patchy and focal similar to that of VCP disease but this is not the case. The more affected muscles show rather diffuse fatty increase without similarity to VCP
- One significant question which needs explanations and comments is why the muscle pathology is similar to the previously reported mutations in this gene despite the opposite effect of the mutations on the fibrillization properties of the mutant protein
- as a consequence of this opposite effect on the fibrillization the explanation of the disease mechanism as caused by a cytoplasmic gain of function effect, which is obvious with the missense mutants, or gain of wt function remains unsatisfactory with loss of normal C-ter as the main problem and would need elaboration.
- similarly the explanation of the hnRNPA2B1 aggregates in the biopsies as final result of abnormal nucleocytoplasmic transport dynamics of frameshift variants, is rather speculative and not supported by findings
- the link to altered autophagic processing as evidenced by the rimmed vacuolar pathology should be clarified

Reviewer #4 (Remarks to the Author):

What are the major claims of the paper?

The authors identified a series of frameshift mutations which occur in the 3' region of the hnRNPA2 locus from 10 independent families with severe early-onset and progressive OPMD condition. At the molecular level, this novel class of frameshift mutations display a distinct type of cellular pathologies. Their findings are novel and provide additional insights into our understanding of OPMD and the structure-function relationship of hnRNPA2B1.

Are they novel and will they be of interest to others in the community and the wider field?

The findings described in the paper are novel and will be of interest mainly to the OPMD community. This work will benefit further from additional mechanistic investigations, which will be of interest to a wider field.

If the conclusions are not original, it would be helpful if you could provide relevant references.

The conclusions are original, but some conclusions need to be further validated.

Is the work convincing, and if not, what further evidence would be required to strengthen the conclusions?

The authors made an initial analysis of a new class of mutations in the hnRNPA2 locus, and observed various cytopathological and biochemical changes in cells and *Drosophila* overexpressed with mutant proteins. Additional functional rescue experiments will help further consolidate the proposed pathogenic mechanism.

On a more subjective note, do you feel that the paper will influence thinking in the field?

The findings presented in the paper will influence thinking in the field. Their work will carry more weight if functional rescue data are provided.

Please feel free to raise any further questions and concerns about the paper.

Figure 1: Patients' consent is not mentioned in the manuscript, particularly for publication of photographs of patients.

In figure 2, the authors presented different levels of microscopic analyses of muscle biopsy of the patients. It is not clear how these microscopic observations, including vacuoles (Fig.2a-h), cytoplasmic/perinuclear/intranuclear tubulofilamentous inclusions (Fig.2i,j), and co-localized P62, ubiquitin 2, TIA1, TDP43 in hnRNPA2B1-positive inclusions (Fig.2k-o) are consequential to the toxicity/cell death described in the cell and fly models. Elucidating the inter-relationships between biopsy observations and the toxicity detected in disease models will allow readers to better understand the pathogenesis.

Figure 3g&h: It is not certain if the synthetic M9-NLS-LCD peptides adopt any secondary structure. If so, how are they different to the wild-type peptide.

Figure 4a&b: The extent of cytoplasmic localization of mutant GFP-hnRNPA2 was not quantified against the wild-type GFP-hnRNPA2 control. It appears that the nucleocytoplasmic distribution of hnRNPA2 (Fig.4b) does not correlate well with cell survival data (Fig. 4d) in the C2C12 model.

The mutant frameshifted hnRNPA2 protein is mislocalized to the cytoplasm. It is uncertain if it is the disrupted Kap-beta-2/hnRNPA2 interaction or other effect(s) of the frameshifts that cause cytotoxicity and in vivo phenotypes in *Drosophila*.

We would also be grateful if you could comment on the appropriateness and validity of any statistical analysis, as well the ability of a researcher to reproduce the work, given the level of detail provided. The data analysis are to the standard.

Minor comment:

Page 5: compositional resemblance to yeast...

We greatly appreciate the reviewers' supportive reception of this report and its conclusions and thank them for their helpful feedback. Our revised manuscript fully addresses all comments, as detailed below. Additional data was generated in response to reviewer suggestions and these are also described below.

To summarize, our manuscript provides detailed clinical characterization of ten independent families with a new form of OPMD-like disease of strikingly early onset (eoOPMD) caused by specific frameshift variants in *hnRNPA2B1*. This clinical syndrome differs from the adult-onset multisystem proteinopathy phenotype caused by missense mutations in *hnRNPA2B1* (Kim et al., 2013).

Further, we describe the molecular consequences of these frameshift mutations, which also differ from the previously described missense mutations in *hnRNPA2B1*, and these findings provide significant insight into the disease mechanism. In contrast to missense mutations in *hnRNPA2B1*, which cause an increased propensity to fibrillization, the frameshift variants result in a neomorphic C-terminal peptide sequence immediately adjacent to the canonical nuclear localization sequence (NLS). These frameshift mutants do not show increased propensity to fibrillize. Rather, the frameshift variants have reduced affinity for the nuclear import receptor Kap β 2, resulting in cytoplasmic accumulation of hnRNPA2 protein in cultured cells and in *Drosophila* models. Despite the lack of an increase in the propensity toward fibrillization, the accumulation of hnRNPA2 in the cytoplasm also culminates in hnRNPA2-positive aggregate pathology and drives cytotoxicity, thereby recapitulating aspects of the human disease and histopathology.

In the peer review process, the reviewers expressed enthusiasm for the impact and quality of this study. They also suggested additional experimental work to extend our understanding of the cellular toxicity and aggregation phenotypes specifically. We therefore designed a series of experiments that provide additional evidence strongly supporting our proposed mechanism of cytotoxicity. These include engineering rescue experiments of the disease-linked hnRNPA2 mutants, designing a series of additional PY-NLS mutants that recapitulate the phenotype, and knockdown experiments that rule out a simple LOF mechanism.

Our work conclusively demonstrates that a class of highly specific heterozygous frameshift variants in *hnRNPA2B1* cause early-onset OPMD. Whereas previously recognized missense mutations associated with increased fibrillization result in adult-onset distal myopathy, the frameshift variants described here result in partial loss of nuclear import. Thus, we conclude that partial loss of NLS activity resulting in cytoplasmic accumulation is the primary disease mechanism for early-onset OPMD.

Reviewers' Comments

Reviewer #1 (Remarks to the Author):

Kim et al have manifestly proved their case that specific heterozygous frameshift variants close to the C-terminus of the protein in *hnRNPA2B1* cause a characteristic myopathy that they have classified as early-onset oculopharyngeal muscular dystrophy. They describe 11 patients in ten families with the phenotype. All have the frameshift variants. In seven of the families, they demonstrate the variants arose de novo. Their functional genomic analyses indicate the pathomechanism of the frameshift variants is different to the pathomechanism of a missense-variants in the gene associated with Paget's disease of bone or multisystem proteinopathy.

There is little room to criticise the manuscript.

Author response: We thank the reviewer for this supportive feedback. We have now addressed all comments in the revised manuscript as described below.

1) One change I would want to see is to include in the Summary an explanation of why the frameshift variants escape nonsense-mediated decay/RNA quality control degradation. This is not included at present. This leaves the reader desperate to know the reason, but they have to read on into the manuscript to obtain the answer.

Please include in the summary that the frameshift variants escape RNA quality control degradation, abolish the stop codon and extend the reading frame.

Author response: This is an excellent suggestion. We have now introduced this sentence early in the Summary so that the reader is not kept in suspense: “The disease-causing frameshift mutations abolish the native stop codon and extend the reading frame, creating a novel transcript that escapes nonsense-mediated decay and is thus translated.”

2) Whether the authors can say that the variants cause the same C-terminal rearrangement could be debated.

Author response: The reviewer is correct. To clarify this point, we now include the following description: “Irrespective of the point at which each frameshift occurred, all mutations cause a shift into the same new frame with a common C-terminal sequence (VMVGGADTELLPICHGLHCINRRG)” (page 10).

3) The authors should also discuss the number of exons in hnRNPA2B1 and which exon(s) the variants are in. For example, are the variants all in the last exon of the gene? Currently there is no discussion in the body of the manuscript of which exon the variants are in. In Supplementary figure 2, there is a diagram showing the variants are in exon 10, which appears to be the last coding exon, with the boundary between exon 10 and exon 11 being immediately after the stop codon. None of the relevance of this to the pathomechanism is discussed in the manuscript, but it should be.

Author response: We appreciate this suggestion and have now added this information to the revised text. To clarify briefly here, hnRNPA2 (NM_002137) and hnRNPB1 (NM_031243) have 11 and 12 exons, respectively. All frameshift variants cluster in exon 10 of hnRNPA2 and exon 11 of hnRNPB1, which are respectively the last coding exon in each isoform. We note that there was a mistake in Supplementary Figure 2a in the boundary between exons 10 and 11 of hnRNPA2, which we have now corrected.

4) It seems remarkable that the frameshift variants spaced so closely together in the gene can be classified according to ACMG guidelines as being pathogenic, likely pathogenic or of uncertain significance (Supplementary Table 1). The reasons for this should be discussed.

Author response: The degree of confidence in the pathogenicity of variants based on ACMG guidelines depends on several factors. These guidelines are conservative in their interpretation of variant pathogenicity and do not depend on the predicted mechanism or effect of the variants on the protein. In this case, all of the variants in our manuscript were initially formally classified as “variants of uncertain significance.” However, when the *de novo* status of a variant was

confirmed, it was possible to re-classify them as “likely pathogenic” on genetic grounds based on ACMG guidelines. In the presence of additional supporting data (e.g., functional data supporting pathogenicity for a given variant), it was then possible to further upgrade the variants to a “pathogenic” classification. We have added corresponding text to the accompanying legend.

5) On page 14 of the PDF, the authors state: “the levels of protein expression were consistently modestly higher in the N323Tfs*36, G328Afs*31, and G331Efs*28 flies compared with the hnRNPA2 WT and D290V flies, suggesting that the frameshift mutations may increase the stability of hnRNPA2 protein in fly muscles”. The explanation given as to why there is increased protein is that the frameshifted protein may be more stable. The explanation is not that the expression is higher. Therefore, please simply state: “the levels of protein were consistently modestly higher ..”

Author response: We agree with the reviewer’s suggestion and have changed this wording to read “protein levels were consistently modestly higher...”

6) The DAPI stain in Figure 6c is odd, it does not encompass the full extent of the anti-hnRNPA2, suggesting that the full extent of the nuclei has not been shown by the DAPI stain. Comment?

Author response: Yes, in our original Figure 6c the gain of the DAPI signal was faint and failed to adequately illustrate the euchromatic regions of the nuclei. In the revised figure (now **Figure 7c**), we have increased the gain of the DAPI signals to show the full extent of the nuclei, demonstrating that hnRNPA2 signal is fully contained within the DAPI-positive nuclei.

Reviewer #2 (Remarks to the Author):

The authors present exciting new results on frameshift variants of hnRNPA2B1 that are here shown to cause oculopharyngeal muscular dystrophy. The case for causation is clear with both genetic linkage and multiple similar variants in families/patients all disrupting the sequence immediately following the PY in the PY NLS of hnRNPA2. These variants are an interesting contrast to the missense variants found in hnRNPA2B1 and hnRNPA1 that cause MSP, suggesting that cytoplasmic localization may be more important than cytoplasmic aggregation to toxicity in cells. Overall the manuscript is exciting. The authors can consider the following questions that could improve the overall findings.

Author response: We appreciate the reviewer’s positive feedback and have addressed each of the reviewer’s questions in the revised manuscript as described below.

More insight in the biophysical mechanism. Does it really prevent aggregation in cells – more clear images showing D290V vs these frameshifts? I think the data are there, but it is somewhat unclear.

Author response: To clarify, we are not stating that the frameshift mutations prevent aggregation in cells, but rather that they do not increase the propensity toward aggregation as found for the previously recognized missense mutations. Indeed, the difference in aggregation behavior of these two classes of mutations is clear and consistent. To more conclusively demonstrate the distribution of hnRNPA2 variant classes in cells, in the revised manuscript we include measurements of signal intensities of WT, the MSP-associated D290V mutant, and

N323Tfs protein in the nucleus and cytoplasm. As shown in **Supplementary Figure 4**, line scan intensity graphs indicate that both WT and D290V predominantly localize to the nucleus before stress (**Supplementary Figure 4a**). Upon oxidative stress (500 μ M sodium arsenite), D290V, but not WT, showed association with stress granules, as demonstrated by colocalization with eIF4G (**Supplementary Figure 4b**). However, the majority of the signal was still found in the nucleus. In contrast, the N323Tfs variant showed substantial accumulation in the cytoplasm before stress (**Supplementary Figure 4a**) and strong association with stress granules upon oxidative stress (**Supplementary Figure 4b**). When not associated with stress granules, the cytoplasmic distribution of N323Tfs was similar to that of eIF4G, which is diffuse in the cytoplasm, demonstrating that N323Tfs protein remains diffuse rather than in aggregates. We also analyzed RIPA solubilities of WT and frameshift mutant proteins. As presented in **Supplementary Figure 4e**, all three frameshift variants were largely recovered from the RIPA-soluble fraction. This stands in contrast to the MSP-associated missense variant D290V, which is mostly recovered from the RIPA-insoluble fraction. All of these results are consistent with the interpretation that the frameshift variants do not increase the propensity towards fibrillization in cells.

Moreover, this interpretation was corroborated *in vivo* in *Drosophila* as shown in **Figure 7c-d** (Figure 6c-d of the original manuscript). In fly muscle tissue, D290V robustly accumulated in cytoplasmic inclusions, whereas all frameshift variants showed diffuse sarcoplasmic signal. Consistent with this finding and our cellular data, protein solubility assays demonstrated that the D290V mutant was RIPA-insoluble, whereas all frameshift variants were RIPA-soluble (**Figure 7e-f**).

More insight in the toxicity mechanism. Is the PY-NLS still functional partially, is a delta PY-NLS toxic more or less, or is it something particular about deletion after the PY that leads to toxicity? What smaller deletions could be made to localize the residues that lead to disruption in binding and increased toxicity? It is somewhat surprising that the region after the PY is so important as several proteins have the final residues as PY (e.g. there is no flanking sequence).

Author response: These are insightful questions that we have now addressed experimentally. To do so, we designed a series of experiments that provide more clear insight into the mechanism of cytotoxicity. These include designing a series of additional PY-NLS mutants that recapitulate the cellular phenotype and engineering rescue experiments of the disease-linked hnRNPA2 mutants. The results of these additional experiments are included in the revised manuscript and are described succinctly here.

Additional PY-NLS mutants included the introduction of missense mutations to key residues in the NLS (P318A and Y319A) or deletions of key residues in the NLS (Δ PY and Δ PY-NLS). We found that single amino acid substitution (P318A or Y319A) or a deletion (Δ PY) of the PY motif did not alter nuclear localization of hnRNPA2 (**Figure 6a, b**). In contrast, the Δ PY-NLS mutant (in which residues spanning the two conserved PY-NLS consensus motifs, **YGPMKSGNFGGSRNMGGPY** [aa 301-319], were deleted) showed cytoplasmic accumulation of the protein that was comparable to the disease-associated N323Tfs*36 variant (**Figure 6a, b**). The Δ 323-341 mutant (in which the C-terminal flanking sequence was deleted) also showed cytoplasmic accumulation of the protein that was comparable to the N323Tfs*36 variant (**Figure 5b, c**). Taken together, these results reveal the importance of PY-NLS flanking sequences in binding Kap β 2. Moreover, in detailed follow-up we have now conducted NMR studies of the interaction between the PY-NLS of hnRNPA2 and Kap β 2 that reveal the binding energy to be distributed across not only the PY dipeptide residues, but also the C-terminal flanking sequence (unpublished). These insights explain why alteration of either the PY-NLS or the C-terminal

flanking sequence causes partial loss of function in Kap β 2 binding. This partial shift to cytoplasmic localization is sufficient to drive cellular toxicity. Indeed, expression of the artificial Δ PY-NLS mutant in differentiated C2C12 cells, which causes modest cytoplasmic accumulation of this protein, results in cellular toxicity similar to the frameshift mutants (**Figure 6c, d**).

To complement these additional PY-NLS mutants and confirm the importance of a functional NLS to prevent cellular toxicity, we designed a rescue experiment. Specifically, we added the canonical monopartite cNLS sequence from simian virus 40 (SV40) T-antigen to the C terminus of N323Tfs*36. Not only did appending the NLS sequence onto this mutant hnRNPA2 fully rescue nuclear localization of N323Tfs*36 (**Figure 6a, b**), but it also rescued cellular toxicity to wild-type levels (**Figure 6c, d**).

The reviewer raises an interesting question about why the mutations downstream of the PY-NLS in hnRNPA2 are so consequential despite the fact that some other members of the PY-NLS family (e.g., FET family members FUS, EWS, and TAF15) do not contain C-terminal flanking sequences. The designation “PY-NLS” refers to nuclear localization signals with a conserved PY-dipeptide motif and these client proteins are typically recognized and transported by the nuclear receptor Kap β 2. Importantly, however, the mode of interaction between Kap β 2 and PY-NLS-containing clients differs from protein to protein. The PY-NLS of hnRNPA1 and hnRNPA2B1 are embedded within the C-terminal low complexity domain and binding to Kap β 2 is supported by flanking sequences. Indeed, structural studies of hnRNPA1 revealed that in addition to modest energetic contributions of the PY residues to binding Kap β 2, important electrostatic interactions are also provided by the adjacent C-terminal flanking regions (Lee et al., 2006). This is confirmed by our own NMR studies of the interaction between hnRNPA2 and Kap β 2 (unpublished).

Based on all of these studies, we conclude that the WT C-terminal flanking sequence is important for nuclear localization of hnRNPA2 by regulating its interaction with the nuclear transport receptor Kap β 2, that this interaction is partly impaired by the disease-causing frameshifting mutations, and that the resulting altered distribution of hnRNPA2 leads to cellular toxicity.

The cell biology data with IF using two different antibodies (custom polyclonal surely fine) that distinguish the C-terminal regions of the proteins. It would help reveal the mechanism to understand if the mutants end up aggregating in the cell lines as well as in patient tissue.

Author response: We agree that developing a tool to distinguish the C-terminal regions of the WT and mutant proteins is a terrific idea. However, the C-terminal domain of WT protein is disordered (now confirmed by our follow-up NMR studies) and for that reason we have had poor luck in generating specific antibodies to these disordered epitopes. Thus, as an alternative approach that is useful in disease models, we introduced fluorescent C-terminal tags to WT and frameshift mutants and characterized their respective distributions. Now presented in **Supplementary Figure 4c**, C-terminal GFP-tagged hnRNPA2 proteins showed equivalent localization patterns as N-terminal GFP-tagged hnRNPA2 proteins (WT in the nucleus, frameshift mutant proteins accumulating in the cytoplasm). We observed no aggregation of C-terminal-tagged frameshift mutant proteins in the cytoplasm of C2C12 cells, consistent with prior observations using N-terminal tagged mutant proteins in HeLa and C2C12 cells (**Figure 3c-f**). The C-terminal GFP-tagged hnRNPA2 proteins migrated on a gel with an observed molecular weight of ~62 kDa, which is close to the estimated molecular weight of full-length protein (hnRNPA2 37 kDa + GFP 27 kDa). No cleaved product was observed by Western blot analyses (**Supplementary Figure 4d**). Moreover, solubility assays revealed that all three frameshift

variants are more RIPA-soluble than WT proteins (**Supplementary Figure 4e**). Thus, we conclude that frameshift mutant proteins are less aggregation-prone compared with WT protein.

Due to the lack of an antibody that distinguishes the C-terminal regions of the WT and mutant proteins, we do not have data addressing the extent to which mutants accumulate in hnRNPA2 aggregates in patient tissue. We do know that these frameshift variants are less prone to fibrillization (**Supplementary Figure 7**), yet it would not be surprising to find these proteins within the complex, amorphous aggregation pathology. These pathological aggregates include multiple proteins, some of which represent fibrillar pathology but also others that are recruited by other valencies, such as protein-protein and RNA-protein interactions. New tools will need to be developed to determine the extent to which the frameshift mutant proteins are recruited to pathological inclusions.

Do the authors think similar variants will emerge in hnRNPA1?

Author response: Yes, indeed we recently reported and characterized four novel hnRNPA1 C-terminal frameshift variants, yielding three protein variants (*321Eext*6, *321Qext*6, G304Nfs*3) that cause ALS and multisystem proteinopathy (MSP) in a broad spectrum of patients with hereditary motor neuropathy, ALS, and myopathy (Beijer et al., 2021). The G304Nfs*3 variant was independently confirmed by another group (Hackman et al., 2021). Intriguingly, the phenotypes associated with *hnRNPA1* frameshift variants are similar to those of *hnRNPA1* missense variants previously identified in ALS and MSP, unlike *hnRNPA2B1* frameshift variants that cause eoOPMD, which have a different phenotype from the missense variants. The basis of phenotypic differences between *hnRNPA1* and *hnRNPA2B1* frameshift variants is not fully understood.

The claim that fibrilization is slowed in vitro is clear in this experiment but it is not clear if this is a robust finding or if it is dependent on particular in vitro conditions that aren't applicable in cells. I suggest mentioning this possibility at least and saying these data "support" but maybe not "strongly suggestive" as the frameshifts also aggregate and in the TEM do not look that qualitatively different from WT.

Author response: We appreciate this comment and in the revised manuscript have used the more conservative language, as suggested.

"Cytoplasmic aggregate formation was associated with the solubility of hnRNPA2" This statement is not clear

Author response: We thank the reviewer for pointing out this awkward turn of phrase. We intended to convey that the missense D290V variant that causes cytoplasmic inclusions is also detergent-insoluble, whereas the frameshift variants described here do not cause cytoplasmic inclusions and remain detergent-soluble. In other words, the histopathological finding of inclusions and the biochemical feature of solubility are correlated, as expected. We have edited the text to clarify this point.

The percentage of the protein in the pellet (Fig S6A) looks qualitatively different for WT and D290V than in the Kim et al 2013 paper where the same assay (I think) was performed. The authors should comment on this.

Author response: Yes, this difference reflects differences in protein concentrations used in the respective experiments. Specifically, in our 2013 paper (Kim et al., 2013) we used 3 μ M

hnRNPA2 and detected fibrillization of WT protein at 8 hours. Under these conditions, we detected no fibrillization of the hnRNPA2 frameshift mutants. Thus, to increase our sensitivity to detecting fibrillization, we increased the concentration of hnRNPA2 to 5 μ M. At this concentration, we detected fibrillization of WT hnRNPA2 at 4 hours, yet still detected no fibrillization of the hnRNPA2 frameshift mutants. The results using 5 μ M are shown in the current manuscript. We have added a brief explanation of these observations to the accompanying figure legend.

Reviewer #3 (Remarks to the Author):

The authors describe a considerable cohort of 10 unrelated families with patients sharing a relatively distinct phenotype consisting of early onset progressive muscular dystrophy with ptosis, ophthalmoparesis and bulbar weakness, caused by different heterozygous frameshift mutations in the PrLd of the MSP gene HNRNPA1. This discovery defines a novel disease phenotype associated with the gene and in contrast with previous mutants these frameshifts seem to reduce the fibrillization properties of the mutants.

Concerns: There are no major concerns with this excellent work

Author response: We thank the reviewer for this supportive feedback. We have now addressed all comments in the revised manuscript as described below.

Comments:

- Muscle MRI findings are described as patchy and focal similar to that of VCP disease but this is not the case. The more affected muscles show rather diffuse fatty increase without similarity to VCP

Author response: We thank the reviewer for pointing this out and welcome the opportunity to clarify our description. Our initial description (*“Presence of patchy foci of T1 signal hyperintensity, suggestive of focal fatty replacement of muscle, was similar to what has been reported for related disorders such as VCP-related MSP”*) referred to more mildly affected muscles. As the reviewer points out, severely affected muscles showed rather diffuse fatty increase. To avoid confusion, we have now revised the manuscript text to describe the pattern of T1 hyperintensity without a direct comparison between VCP-related disease and the MRI of our present patients. The revised description reads as follows: *“Presence of patchy foci of T1 signal hyperintensity, suggestive of focal fatty replacement of muscle, was noted in mildly affected muscles, while more severely affected muscles showed a diffuse pattern of T1 hyperintensity.”*

- One significant question which needs explanations and comments is why the muscle pathology is similar to the previously reported mutations in this gene despite the opposite effect of the mutations on the fibrillization properties of the mutant protein

Author response: We appreciate the opportunity to expand upon and clarify this important point. As the reviewer points out, the two classes of mutations have different consequences for the hnRNPA2 protein fibrilization; however, we speculate that the long-term consequence of both classes of mutations is the accrual of amorphous aggregates that include the hnRNPA2 protein. Our working model is that there are two distinct, but not mutually exclusive mechanisms that can lead to pathological proteinaceous deposits (Mathieu et al., 2020; Nedelsky and Taylor, 2021). First, pathological aggregates can arise from mutations that reduce the energy threshold

for fibrillization. In this category are mutations that alter key residues in the low complexity domain of RNA-binding proteins (e.g., D290V in hnRNPA2B1, A315T in TDP-43). Second, pathological aggregates can also arise through the evolution of poorly dynamic RNP granules. RNP granules are dynamic liquids that, when perturbed, can undergo a liquid-to-solid phase transition that culminates in pathological aggregates similar to those that arise through primary nucleation. Such perturbations include increased concentration of constituent proteins (e.g., RNA-binding proteins) and impairment of regulatory factors (e.g., VCP). Disease-causing mutations in this second category include (1) NLS mutations that increase the cytosolic concentration of RNA-binding proteins (e.g., the frameshift mutations in hnRNPA2B1 reported here, and a series of mutations in FUS that are clustered at the NLS), (2) hexanucleotide expansion in *C9ORF72* that produces dipeptide repeats that insinuate into biomolecular condensates and alter their material properties (Boeynaems et al., 2017; Lee et al., 2016), and (3) mutations in VCP that impair its ability to disassemble RNP granules (Gwon et al., 2021). Although the nature and direct molecular consequences of different classes of mutation are clearly different (e.g., D290V vs. the eoOPMD frameshift variants in *hnRNPA2B1*), mutations in both categories can ultimately lead to pathological phase transition and deposition of histologically evident protein inclusions over time. We have clarified this point in the revised manuscript.

- as a consequence of this opposite effect on the fibrillization the explanation of the disease mechanism as caused by a cytoplasmic gain of function effect, which is obvious with the missense mutants, or gain of wt function remains unsatisfactory with loss of normal C-ter as the main problem and would need elaboration.

Author response: As the reviewer points out, the missense mutations likely drive primary nucleation of hnRNPA2 fibrils via a toxic gain of function mechanism. In contrast, the second class of mutations described here partially impair nuclear import – thus a partial loss of function. However, one of the main consequences of this impaired nuclear import is persistently increased concentration of hnRNPA2 in the cytoplasm. Thus, we suggest that the resulting increased cytoplasmic concentration ultimately results in a gain of toxic function in the cytoplasm. The exact mechanistic consequences resulting in toxicity may still be different for these two classes of mutation and remain to be fully elucidated. Taken together, the concurrent impaired nuclear import in addition to cytoplasmic accumulation of hnRNPA2 distinctly compromise the normal hnRNPA2 subcellular distribution dynamics, in contrast to the missense variants. Supporting this model, our knockdown experiments included in the revised manuscript show that depletion of endogenous *hnRNPA2B1* by siRNA does not cause toxicity in C2C12 myoblasts, arguing against a simple LOF mechanism (**Supplementary Fig. 5a-c**).

- similarly the explanation of the hnRNPA2B1 aggregates in the biopsies as final result of abnormal nucleocytoplasmic transport dynamics of frameshift variants, is rather speculative and not supported by findings

Author response: We agree with the reviewer that this explanation is speculative. Our hypothesis is based on our growing understanding from studies in related molecules. For example, the majority of familial ALS-associated FUS mutations occur within the NLS and impair nuclear import of FUS (Dormann et al., 2010; Gal et al., 2011). In cells, these mutant proteins redistribute to the cytoplasm and associate with stress granules. In patient tissues, these mutant proteins and stress granule markers co-deposit into proteinaceous inclusions, implicating stress granule formation in the pathogenesis of ALS. In a transgenic animal model, expression of NLS mutant FUS triggers proteinopathy and a severe motor phenotype

(Shelkovernikova et al., 2013). Thus, we suggest a similar progression for frameshift *hnRNPA2B1* variants in skeletal muscle.

Furthermore, as described above, liquid condensates such as RNP granules host the nucleation of fibrils (Molliex et al., 2015; Ray et al., 2020; Safari et al., 2019). Based on this circumstantial evidence, we speculate that nucleocytoplasmic transport defects that trigger an increase in the concentration of RNP granule constituent proteins (e.g., FUS, hnRNPA1, hnRNPA2B1) result in proteins crossing their saturation threshold concentration, undergoing LLPS, and assembling into RNP granules that become poorly dynamic, ultimately facilitating the formation of fibrils that are found in patient tissues.

- the link to altered autophagic processing as evidenced by the rimmed vacuolar pathology should be clarified

Author response: Indeed, we did observe pathological evidence of rimmed vacuoles that appear to be of autophagic origin in the muscle biopsy samples. However, the relevance of this nonspecific finding to the pathophysiology of the observed myopathic phenotype remains unclear and could reflect an epiphenomenon/end-stage pathologic manifestation of multiple disease processes. To reflect this position, we have added the following text to the discussion section: *“The relevance of autophagic vacuoles and tubular filamentous inclusions to the pathophysiology of the specific myopathic phenotype observed in our patients remains unclear.”*

Reviewer #4 (Remarks to the Author):

What are the major claims of the paper?

The authors identified a series of frameshift mutations which occur in the 3' region of the *hnRNPA2* locus from 10 independent families with severe early-onset and progressive OPMD condition. At the molecular level, this novel class of frameshift mutations display a distinct type of cellular pathologies. Their findings are novel and provide additional insights into our understanding of OPMD and the structure-function relationship of *hnRNPA2B1*.

Are they novel and will they be of interest to others in the community and the wider field?

The findings described in the paper are novel and will be of interest mainly to the OPMD community. This work will benefit further from additional mechanistic investigations, which will be of interest to a wider field.

If the conclusions are not original, it would be helpful if you could provide relevant references.

The conclusions are original, but some conclusions need to be further validated.

Is the work convincing, and if not, what further evidence would be required to strengthen the conclusions?

The authors made an initial analysis of a new class of mutations in the *hnRNPA2* locus, and observed various cytopathological and biochemical changes in cells and *Drosophila* overexpressed with mutant proteins. Additional functional rescue experiments will help further consolidate the proposed pathogenic mechanism.

On a more subjective note, do you feel that the paper will influence thinking in the field?

The findings presented in the paper will influence thinking in the field. Their work will carry more weight if functional rescue data are provided.

We would also be grateful if you could comment on the appropriateness and validity of any statistical analysis, as well the ability of a researcher to reproduce the work, given the level of detail provided.

The data analysis are to the standard.

Please feel free to raise any further questions and concerns about the paper.

Author response: We thank the reviewer for this supportive feedback. As suggested, we designed a series of experiments that provide more clear insight into the mechanism of cytotoxicity. These include designing a series of additional PY-NLS mutants that recapitulate the cellular phenotype and engineering rescue experiments of the disease-linked hnRNPA2 mutants. The results of these additional experiments are included in the revised manuscript and are described succinctly here.

Additional PY-NLS mutants included the introduction of missense mutations to key residues in the NLS (P318A and Y319A) or deletions of key residues in the NLS (Δ PY and Δ PY-NLS). We found that single amino acid substitution (P318A or Y319A) or a deletion (Δ PY) of the PY motif did not alter nuclear localization of hnRNPA2 (**Figure 6a, b**). In contrast, the Δ PY-NLS mutant (in which residues spanning the two conserved PY-NLS consensus motifs, **YGPMKSGNFGGSRNMGGPY** [aa 301-319], were deleted) showed cytoplasmic accumulation of the protein that was comparable to the disease-associated N323Tfs*36 variant (**Figure 6a, b**). The Δ 323-341 mutant (in which the C-terminal flanking sequence was deleted) also showed cytoplasmic accumulation of the protein that was comparable to the N323Tfs*36 variant (**Figure 5b, c**). Taken together, these results reveal the importance of PY-NLS flanking sequences in binding Kap β 2. Moreover, in detailed follow-up we have now conducted NMR studies of the interaction between the PY-NLS of hnRNPA2 and Kap β 2 that reveal the binding energy to be distributed across not only the PY dipeptide residues, but also the C-terminal flanking sequence (unpublished). These insights explain why alteration of either the PY-NLS or the C-terminal flanking sequence causes partial loss of function in Kap β 2 binding. This partial shift to cytoplasmic localization is sufficient to drive cellular toxicity. Indeed, expression of the artificial Δ PY-NLS mutant in differentiated C2C12 cells, which causes modest cytoplasmic accumulation of this protein, results in cellular toxicity similar to the frameshift mutants (**Figure 6c, d**).

To complement these additional PY-NLS mutants and confirm the importance of a functional NLS to prevent cellular toxicity, we designed a rescue experiment. Specifically, we added the canonical monopartite cNLS sequence from simian virus 40 (SV40) T-antigen to the C terminus of N323Tfs*36. Not only did appending the NLS sequence onto this mutant hnRNPA2 fully rescue nuclear localization of N323Tfs*36 (**Figure 6a, b**), but it also rescued cellular toxicity to wild-type levels (**Figure 6c, d**).

Based on all of these studies, we conclude that the WT C-terminal flanking sequence is important for nuclear localization of hnRNPA2 by regulating its interaction with the nuclear transport receptor Kap β 2, that this interaction is partly impaired by the disease-causing frameshifting mutations, and that the resulting altered distribution of hnRNPA2 leads to cellular toxicity.

Figure 1: Patients' consent is not mentioned in the manuscript, particularly for publication of photographs of patients.

Author response: Patient consent was obtained for these research studies, including consent to publish patient photos. The details of this consent and relevant IRB protocols are listed in the online methods and have now been amended to mention specific consent for publishing photos. The photos of the two patients shown in the manuscript were cleared by the patients and/or guardians.

In figure 2, the authors presented different levels of microscopic analyses of muscle biopsy of the patients. It is not clear how these microscopic observations, including vacuoles (Fig.2a-h), cytoplasmic/perinuclear/intranuclear tubulofilamentous inclusions (Fig.2i,j), and co-localized P62, ubiquitin 2, TIA1, TDP43 in hnRNPA2B1-positive inclusions (Fig.2k-o) are consequential to the toxicity/cell death described in the cell and fly models. Elucidating the inter-relationships between biopsy observations and the toxicity detected in disease models will allow readers to better understand the pathogenesis.

Author response: Elucidating the relationship between histopathological findings and the molecular bases of neurological and muscle diseases is an important but challenging issue that has been gradually illuminated through decades of research. The specific histopathological features of eoOPMD described here overlap with features of other myopathies and, indeed, with some neurodegenerative diseases, suggesting some overlap in the underlying disease mechanisms or at least a similar final pathway. Vacuolization is a frequent histopathological feature in muscle diseases that, while difficult to assign specificity, may reflect hyperactivity of autophagy or, alternatively, defects in completion of autophagy. Primary disease-causing mutations in certain proteins enriched in inclusions/aggregates, including RNA-binding proteins (e.g., hnRNPA2B1, hnRNPA1, TDP-43 and TIA1) and ubiquitin-binding proteins (e.g., p62 and UBQLN2) have been reported. However, deposits, inclusions and aggregates have also been reported outside of this group of disorders and may be found in late-onset myopathies, inclusion body myositis, MSP, and sporadic neurodegenerative diseases such as ALS/FTD.

The hypothesis that the proteinaceous deposits play a role in the pathogenesis of the disease is further supported by the observation that the exogenous expression of variant proteins based on the disease-causing mutations in two simple model systems (e.g., cell culture and *Drosophila*) recapitulates aspects of the disease phenotype, including cellular toxicity and degeneration. We however also realize that there are important caveats, since human diseases evolve over years, whereas the degenerative phenotypes modeled in cell culture and *Drosophila* evolve over days or weeks.

As described above in response to Reviewer 3, our working model is that, although the nature and direct molecular consequences of different classes of mutation are clearly different (e.g., D290V vs. the eoOPMD frameshift variants in hnRNPA2B1 fibrillization propensity), mutations in both categories can ultimately lead to pathological phase transition and deposition of histopathologically evident protein inclusions over time (see above for a more in-depth discussion). Because RNP granules are composed of numerous RBPs, changes in the material properties of these RNP granules can lead to co-deposition of many RBPs (e.g., TIA1, TDP-43, and hnRNPA2B1) in the inclusion. Failed cellular attempts to eliminate these abnormal protein aggregates by the ubiquitin-proteasome pathway can also leave co-deposition of proteins involved in this pathway (e.g., ubiquitin, UBQLN2, SQSTM1/p62, and VCP). We have edited the text of the manuscript to better communicate this model.

Figure 3g&h: It is not certain if the synthetic M9-NLS-LCD peptides adopt any secondary structure. If so, how are they different to the wild-type peptide.

Figure 1. Secondary structure of WT and N323Tfs*36 C-terminal peptides predicted using the SSP algorithm. Both WT and frameshifted peptides have narrow proton NMR chemical shift dispersion, which is characteristic of no persistent structure. Secondary structural propensities calculated from $C\alpha$ and $C\beta$ chemical shifts are identical between the two peptides where the sequences overlap. The frameshifted variant has a higher propensity for helical conformations in the C terminus compared to WT but remains disordered.

Author response: This is an important point that we have now also addressed experimentally. We performed an NMR spectroscopy study and determined the secondary structure of C-terminal peptides (WT and frameshift variant) from the chemical shift information. As shown in Figure 1 above, both WT and frameshifted peptides have narrow proton NMR chemical shift dispersion, which is characteristic of no persistent structure. The frameshifted variant has a higher propensity for helical conformations in the C terminus compared to WT, but remains disordered. These details are beyond the scope of the current manuscript but are fully explored in a forthcoming paper that explores the structural basis of hnRNPA2 binding to Kap β 2 and the consequences of various mutations.

Figure 4a&b: The extent of cytoplasmic localization of mutant GFP-hnRNPA2 was not quantified against the wild-type GFP-hnRNPA2 control. It appears that the nucleocytoplasmic distribution of hnRNPA2 (Fig.4b) does not correlate well with cell survival data (Fig. 4d) in the C2C12 model.

Author response: This is an essential point and we welcome the opportunity to resolve the apparent contradiction. In our revised manuscript we present quantitative data that supports our initial conclusion that nucleocytoplasmic localization of hnRNPA2 correlates with cell survival data. As shown in revised **Figure 3f**, the frameshift mutations in GFP-tagged hnRNPA2 show altered nucleocytoplasmic ratios of proteins (8%, 25%, 23%, and 22% cytoplasmic accumulation in WT, N323Tfs*36, G328Afs*31, and G331Efs*28 variants, respectively, reflecting ~3-fold increased cytoplasmic accumulation for the frameshift variants), suggesting a nuclear import defect for the frameshift variants. In cell toxicity assays, expression of WT hnRNPA2 caused ~40% and ~60% cell toxicity at day 1 and day 2 post differentiation, and expression of frameshift variants caused ~60% and ~80% cell toxicity at day 1 and day 2 post-differentiation, demonstrating ~1.5-fold increased cell toxicity for the frameshift variants (**Figure 4d**, **Figure 5e**, and **Figure 6d**). Thus, the nuclear import defects of the frameshift variants correlate well with cell toxicity.

The mutant frameshifted hnRNPA2 protein is mislocalized to the cytoplasm. It is uncertain if it is the disrupted Kap-beta-2/hnRNPA2 interaction or other effect(s) of the frameshifts that cause cytotoxicity and in vivo phenotypes in *Drosophila*.

Author response: We thank the reviewer for giving us the opportunity to more clearly present our model linking cytotoxicity to the cytoplasmic accumulation of hnRNPA2 protein. As described above in our first response to this reviewer, our revised manuscript includes new lines of evidence supporting our proposed pathological mechanism, including rescue experiments of the disease-linked hnRNPA2 mutants and testing of additional PY-NLS mutants that recapitulate the phenotype.

Our working model suggests that mutations that impair nuclear import, and consequently cause persistently increased concentrations of hnRNPA2 in the cytoplasm, result in a toxic gain of function in the cytoplasm. Substantial evidence has now accrued that increased concentration of an RNP granule constituent is one of several mechanisms that impairs the dynamics of RNP granules, culminating in liquid-to-solid phase transition and evolution toward pathological aggregates that also include hnRNPA2 protein (Mathieu et al., 2020; Nedelsky and Taylor, 2021). Thus, whereas the primary consequence of the frameshift mutations is impaired binding to Kap β 2, we suggest that the resulting increased cytoplasmic concentration ultimately results in similar consequences as the missense mutations – a toxic gain of function in the cytoplasm. This model is further supported by the results of knockdown experiments that we have included in the revised manuscript. Specifically, we showed that depletion of endogenous *hnRNPA2B1* by siRNA does not cause toxicity in differentiated C2C12 cells, arguing against a simple LOF mechanism (**Supplementary Fig. 5a-c**).

We further note that a simple deletion of C-terminal flanking sequence (Δ 323-341) or +2 frameshift (N323Lfs*31), which causes a sequence rearrangement that differs from the eoOPMD-associated frameshift variants, also cause nucleocytoplasmic transport defects and cellular toxicity in differentiated myoblasts at levels comparable to the eoOPMD-associated frameshift variants (**Figure 5, Supplementary Fig. 5**). Taken together, these results suggest that loss of the wild-type C-terminal flanking sequence, rather than any specific features of the new amino acid sequence induced by the frameshift variants, is responsible for the observed cellular phenotypes associated with the *hnRNPA2B1* frameshift variants.

Minor comment:

Page 5: compositional resemblance to yeast...

Author response: We could not find this error in our manuscript.

References

- Beijer, D., Kim, H.J., Guo, L., O'Donovan, K., Mademan, I., Deconinck, T., Van Schil, K., Fare, C.M., Drake, L.E., Ford, A.F., *et al.* (2021). Characterization of HNRNPA1 mutations defines diversity in pathogenic mechanisms and clinical presentation. *JCI Insight* 6.
- Boeynaems, S., Bogaert, E., Kovacs, D., Konijnenberg, A., Timmerman, E., Volkov, A., Guharoy, M., De Decker, M., Jaspers, T., Ryan, V.H., *et al.* (2017). Phase Separation of C9orf72 Dipeptide Repeats Perturbs Stress Granule Dynamics. *Mol Cell* 65, 1044-1055 e1045.
- Dormann, D., Rodde, R., Edbauer, D., Bentmann, E., Fischer, I., Hruscha, A., Than, M.E., Mackenzie, I.R., Capell, A., Schmid, B., *et al.* (2010). ALS-associated fused in sarcoma (FUS) mutations disrupt Transportin-mediated nuclear import. *EMBO J* 29, 2841-2857.
- Gal, J., Zhang, J., Kwinter, D.M., Zhai, J., Jia, H., Jia, J., and Zhu, H. (2011). Nuclear localization sequence of FUS and induction of stress granules by ALS mutants. *Neurobiol Aging* 32, 2323 e2327-2340.

Gwon, Y., Maxwell, B.A., Kolaitis, R.M., Zhang, P., Kim, H.J., and Taylor, J.P. (2021). Ubiquitination of G3BP1 mediates stress granule disassembly in a context-specific manner. *Science*, In press.

Hackman, P., Rusanen, S.M., Johari, M., Vihola, A., Jonson, P.H., Sarparanta, J., Donner, K., Lahermo, P., Koivunen, S., Luque, H., *et al.* (2021). Dominant Distal Myopathy 3 (MPD3) Caused by a Deletion in the HNRNPA1 Gene. *Neurol Genet* 7, e632.

Kim, H.J., Kim, N.C., Wang, Y.D., Scarborough, E.A., Moore, J., Diaz, Z., MacLea, K.S., Freibaum, B., Li, S., Molliex, A., *et al.* (2013). Mutations in prion-like domains in hnRNPA2B1 and hnRNPA1 cause multisystem proteinopathy and ALS. *Nature* 495, 467-473.

Lee, B.J., Cansizoglu, A.E., Suel, K.E., Louis, T.H., Zhang, Z., and Chook, Y.M. (2006). Rules for nuclear localization sequence recognition by karyopherin beta 2. *Cell* 126, 543-558.

Lee, K.H., Zhang, P., Kim, H.J., Mitrea, D.M., Sarkar, M., Freibaum, B.D., Cika, J., Coughlin, M., Messing, J., Molliex, A., *et al.* (2016). C9orf72 Dipeptide Repeats Impair the Assembly, Dynamics, and Function of Membrane-Less Organelles. *Cell* 167, 774-788 e717.

Mathieu, C., Pappu, R.V., and Taylor, J.P. (2020). Beyond aggregation: Pathological phase transitions in neurodegenerative disease. *Science* 370, 56-60.

Molliex, A., Temirov, J., Lee, J., Coughlin, M., Kanagaraj, A.P., Kim, H.J., Mittag, T., and Taylor, J.P. (2015). Phase separation by low complexity domains promotes stress granule assembly and drives pathological fibrillization. *Cell* 163, 123-133.

Nedelsky, N.B., and Taylor, J.P. (2021). Pathological phase transitions in ALS-FTD impair dynamic RNA-protein granules. *RNA*.

Ray, S., Singh, N., Kumar, R., Patel, K., Pandey, S., Datta, D., Mahato, J., Panigrahi, R., Navalkar, A., Mehra, S., *et al.* (2020). alpha-Synuclein aggregation nucleates through liquid-liquid phase separation. *Nat Chem*.

Safari, M.S., Wang, Z., Tailor, K., Kolomeisky, A.B., Conrad, J.C., and Vekilov, P.G. (2019). Anomalous Dense Liquid Condensates Host the Nucleation of Tumor Suppressor p53 Fibrils. *iScience* 12, 342-355.

Shelkownikova, T.A., Peters, O.M., Deykin, A.V., Connor-Robson, N., Robinson, H., Ustyugov, A.A., Bachurin, S.O., Ermolkevich, T.G., Goldman, I.L., Sadchikova, E.R., *et al.* (2013). Fused in sarcoma (FUS) protein lacking nuclear localization signal (NLS) and major RNA binding motifs triggers proteinopathy and severe motor phenotype in transgenic mice. *J Biol Chem* 288, 25266-25274.

REVIEWERS' COMMENTS

Reviewer #1 (Remarks to the Author):

The authors have satisfactorily addressed all my concerns with the manuscript.

This with their responses to the questions of the other reviewers has, I consider, resulted in an outstanding manuscript.

Reviewer #2 (Remarks to the Author):

The authors should be applauded for their fantastic paper.

Reviewer #3 (Remarks to the Author):

the authors have addressed my concerns

Reviewer #4 (Remarks to the Author):

The authors have fully addressed the concerns I have raised. This work is suitable for publication.

The reviewers were all satisfied with our revised manuscripts and had no further comments. We greatly appreciate the reviewers' supportive reception of our manuscript and thank them for their helpful feedback.

Reviewers' Comments

Reviewer #1 (Remarks to the Author):

The authors have satisfactorily addressed all my concerns with the manuscript. This with their responses to the questions of the other reviewers has, I consider, resulted in an outstanding manuscript.

Reviewer #2 (Remarks to the Author):

The authors should be applauded for their fantastic paper.

Reviewer #3 (Remarks to the Author):

the authors have addressed my concerns

Reviewer #4 (Remarks to the Author):

The authors have fully addressed the concerns I have raised. This work is suitable for publication.